# Neodymium isotope evidence for glacial-interglacial variability of deepwater transit time in the Pacific Ocean

Rong Hu [1,2] & Alexander M. Piotrowski[2]

There is evidence for greater carbon storage in the glacial deep Pacific, but it is uncertain whether it was caused by changes in ventilation, circulation, and biological productivity. The spatial $\varepsilon_{Nd}$ evolution in the deep Pacific provides information on the deepwater transit time. Seven new foraminiferal $\varepsilon_{Nd}$ records are presented to systematically constrain glacial to interglacial changes in deep Pacific overturning and two different $\varepsilon_{Nd}$ evolution regimes occur spatially in the Pacific with reduced meridional $\varepsilon_{Nd}$ gradients in glacials, suggesting a faster deep Pacific overturning circulation. This implies that greater glacial carbon storage due to sluggish circulation, that is believed to have occurred in the deep Atlantic, did not operate in a similar manner in the Pacific Ocean. Other mechanisms such as increased biological pump efficiency and poor high latitude air-sea exchange could be responsible for increased carbon storage in the glacial Pacific.

[1] School of Geography and Ocean Science, Nanjing University, 210023 Nanjing, China. [2] Department of Earth Sciences, University of Cambridge, Cambridge CB2 3EQ, UK. Correspondence and requests for materials should be addressed to R.H. (email: ronghu@nju.edu.cn)

Being volumetrically the largest ocean basin and most enriched in nutrients, Pacific deep water stores the largest amount of dissolved inorganic carbon in the Earth's ocean-atmosphere system today and may have been an even larger carbon reservoir during the glacial periods[1–3]. One of the central questions is how the Pacific Ocean circulation linked to atmospheric $CO_2$ concentration over glacial–interglacial (G–I) cycles. Previous studies in the Atlantic Ocean have attributed the enhanced respired carbon storage at the Last Glacial Maximum (LGM) to an increase in the residence time of deep Atlantic waters[4,5], but few studies have focused on the past variability of Pacific deep ocean circulation and the results are still controversial[3,6–11]. One of the reasons is that foraminifera tests, a primary archive of past ocean chemistry, are poorly preserved in the deep Pacific due to the presence of more corrosive deep waters. The partial dissolution of foraminifera calcite might complicate the reliable reconstruction of foraminifera-based geochemical proxies (such as $\delta^{18}O$, $\delta^{13}C$, $\Delta^{14}C$, Cd/Ca), which otherwise would provide valuable information about the past nutrient and ventilation states of the deep oceans. These proxies are also controlled by a combination of high latitude air-sea exchange, biological production, and ocean circulation, and thus have difficulty in distinguishing whether the increased $CO_2$ storage in the glacial deep Pacific was the result of reduced air-sea exchange in the Southern Ocean or sluggish deepwater circulation.

Neodymium isotope has been demonstrated to be an effective proxy to trace the mixing in the global deep ocean[12–14]. However, unlike north–south convection structure in the Atlantic, there is no deepwater formation in the North Pacific today, and the deep Pacific basin is exclusively ventilated by northward-flowing Lower Circumpolar Deep Water (LCDW). The strong linear relationships between seawater Nd isotope compositions and oxygen/phosphate concentrations in the deep Pacific[12,15] support the dominant role of LCDW advection and the presence of an external (i.e. not from a particular water mass) radiogenic Nd source which is added to the deep ocean in a nearly spatially homogenous way similar to nutrients, but not a link with the biological processes per se[15]. In this regard, the accumulation of external Nd input to the deep water might be time-dependant and the large-scale spatial Nd isotope gradients are likely to be associated with the deep Pacific circulation rate.

To better constrain the temporal and spatial Nd isotope evolution in the deep Pacific, Nd isotope work on seven sediment cores within a range of water depth between 600 and 4000 m from the southwest Pacific (SWP) and eastern/western equatorial Pacific (EEP/WEP) over the last and penultimate glacial maximum (30 and 160 ka) are presented in this study and compared with another 10 $\varepsilon_{Nd}$ records throughout the deep Pacific (Fig. 1 and Supplementary Table 1). Although different archives have been applied to extract seawater Nd isotope signatures in paleoceanographic reconstructions[12], to avoid possible operational bias for the spatial $\varepsilon_{Nd}$ gradients, the $\varepsilon_{Nd}$ records investigated and compiled here are processed under a uniform technique, i.e. foraminifera with Fe-Mn coatings[13], which is demonstrated to reliably record the Nd isotope signatures of porewater/bottom water in the Pacific[12,15]. On this basis, the detrital influence on porewater $\varepsilon_{Nd}$ is examined and controlling factors of Nd isotope evolution from different regions and different depths are summarised. Building on the hypothesis of time-dependant accumulation of external Nd on the LCDW pathway, a simple box model is used to quantitatively estimate the transit time over the past 160 ka. These results will have important implications on how the strength of the overturning circulation is linked to carbon storage in the deep Pacific Ocean over G–I cycles.

## Results

**Deglacial $\varepsilon_{Nd}$ records in the Equatorial Pacific (0–30 ka).** Two Nd isotopic records spanning the last 30 ka were measured on the EEP intermediate-depth core V21-30 and deep WEP core V28-239 (Fig. 2 and Supplementary Tables 2–3). Both foraminiferal and detrital Nd isotopic records of V21-30 are presented. The foraminiferal $\varepsilon_{Nd}$ record shows a continually increasing trend from glacial values $-1.6 \pm 0.2$ at 24.8 ka to $+0.1 \pm 0.2$ at 6.4 ka, and then the Nd isotope values start to stabilize at ~$-1$ to ~0 in the late Holocene. However, the detrital Nd isotope compositions are much more radiogenic (~0 to $+4.5$) and evolved differently from the foraminiferal record. The largest radiogenic shift ($+2.7$ $\varepsilon_{Nd}$ units) in detrital Nd isotopes occurred from 12.4 ka to 10.9 ka, followed by a steadily $\varepsilon_{Nd}$ decreasing trend during the Holocene, when the foraminiferal $\varepsilon_{Nd}$ was in a continuous increasing evolution. Another discrepancy in these two records can be seen during the last glacial period when there was a rapid detrital $\varepsilon_{Nd}$ elevation from $+0.6 \pm 0.2$ at 28.2 ka to $+1.8 \pm 0.2$ at 27.2 ka, followed by a sharp decline to $+0.2 \pm 0.2$ at 26.7 ka, while the foraminiferal Nd isotope compositions showed a negative shift from $-0.9 \pm 0.2$ to $-1.4 \pm 0.2$. On the other hand, the foraminiferal Nd isotope compositions of the western equatorial core V28-239 are less radiogenic and showed little variability ($-3.0$ to $-3.6$, average $\varepsilon_{Nd} = -3.4 \pm 0.2$) across the last 30 kyr (Fig. 2d).

**Nd isotopic records spanning the last glacial (0–160 ka).** Four foraminiferal Nd isotopic records over the last 160 ka are measured on two intermediate SWP sites (SO136-38 and CHAT16K) and two deep EEP sites (ODP1241 and ODP846), along with eight $\varepsilon_{Nd}$ data points from RC13-114 due to limited sample availability for this core (Fig. 3 and Supplementary Tables 4–8). The Nd isotope compositions of the four foraminiferal records vary with core locations, with more radiogenic $\varepsilon_{Nd}$ values farther north. For example, the most southerly core SO136-38 on Campbell Plateau had the least radiogenic $\varepsilon_{Nd}$ values in this study, which varied between ~$-7$ during interglacial times (MIS 1 and 5) and ~$-5.5$ during glacial periods (MIS 2 and 6). To the north of Chatham Rise, the intermediate-depth core CHAT16K had much more radiogenic $\varepsilon_{Nd}$ values (average $= -3.6 \pm 0.3$) and less variability than SO136-38. The $\varepsilon_{Nd}$ values of ODP846 in the deep EEP also did not vary much over the past 160 ka and were generally about 1 $\varepsilon_{Nd}$ unit higher than CHAT16K, with an average $\varepsilon_{Nd}$ value of $-2.6 \pm 0.4$. Although there are only eight $\varepsilon_{Nd}$ data points for RC13-114 over the past 160 ka, these $\varepsilon_{Nd}$ data (average $= -2.6 \pm 0.4$) match well with the nearby record of ODP846, confirming that they are recording the $\varepsilon_{Nd}$ signals of the same water mass. The most northern site ODP1241 in the EEP showed the most radiogenic foraminiferal $\varepsilon_{Nd}$ values in this study, ranging between $-1.5$ and $+1$, with less radiogenic $\varepsilon_{Nd}$ signatures during glacial periods (MIS 2 and 6). Unlike the foraminiferal Nd isotope record, the Nd isotopes of the detrital fractions did not change following G–I cycles. Instead, the least radiogenic detrital Nd isotope composition ($-2.1 \pm 0.2$) occurred at 135 ka, whereas the most radiogenic signal appeared at late Holocene ($+4.4 \pm 0.2$) and MIS 4 ($+3.3 \pm 0.1$).

## Discussion

The detrital $\varepsilon_{Nd}$ values of V21-30 varied between 0 and $+5$ (Fig. 2c) and those of ODP1241 varied between $-3$ and $+5$ (Fig. 3c) during the studied time periods, indicating a dominant proximal source of central American arcs/Galapagos hotspot to this region[16,17]. The consistency of decreased detrital $\varepsilon_{Nd}$ in the equatorial Pacific[18–20] during the glacial periods is in accordance with globally higher continental dust loads[21] from Asia, South

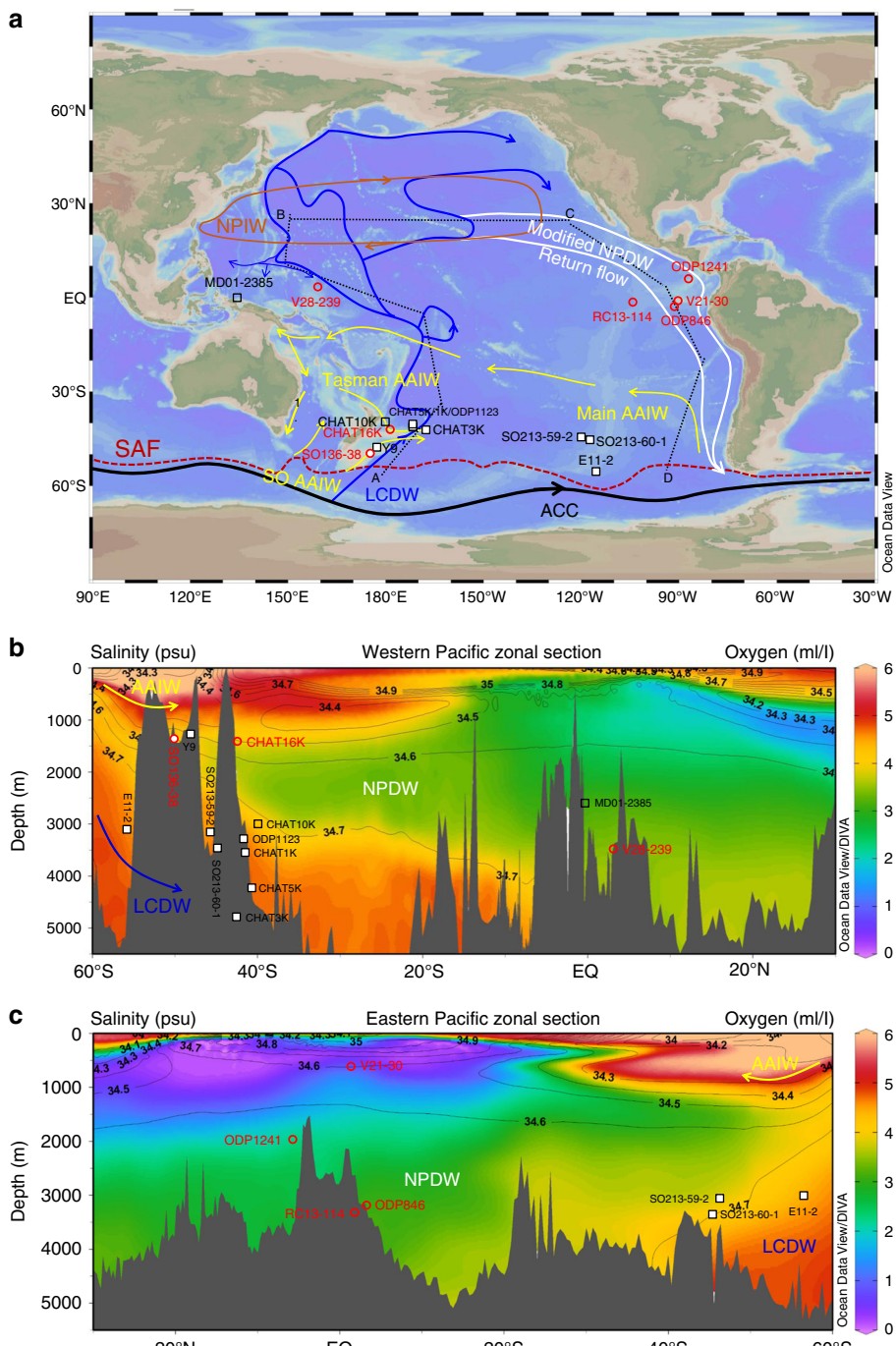

**Fig. 1** Modern hydrography and core locations in the Pacific Ocean. **a** Map showing intermediate and deep water circulation and locations of the sediment cores from this study (red circles) and other Pacific sites (black squares) for comparison (Site locations and sources please refer to Supplementary Table 1). White filled symbols denote cores showing South Pacific-regime Nd isotope evolution patterns, while the open symbols represent cores showing North Pacific-regime patterns (see text). SAF: Subantarctic Front, ACC: Antarctic Circumpolar Current, LCDW: Lower Circumpolar Deep Water, AAIW: Antarctic Intermediate Water, NPDW: North Pacific Deep Water, NPIW: North Pacific Intermediate Water. **b** Western Pacific zonal section of dissolved oxygen (colour) and salinity (contours) along the dark grey dotted line from A to B in Map **a**. **c** Eastern Pacific zonal section of dissolved oxygen (colour) and salinity (contours) along the dark grey dotted line from C to D in Map **a**. Note that the three cores from the Central South Pacific are included in both sections solely on their latitudes and depths. Figures created using Ocean Data View Software (http://odv.awi.de)

America and/or Africa. From the perspective of Nd isotopes, the detrital sources of our EEP cores can thus be divided into two endmembers: one from young volcanic arcs with radiogenic $\varepsilon_{Nd}$ signatures (average $\varepsilon_{Nd} = +7$)[22] and the other from old continental particles with unradiogenic $\varepsilon_{Nd}$ signatures (average

$\varepsilon_{Nd} = -10$)[23]. The variation in detrital $\varepsilon_{Nd}$ records of V21-30 and ODP1241 can be regarded as reflecting changing proportions of inputs from these two endmembers.

The foraminiferal Nd isotopic composition of the youngest material of V28-239 ($-3.3 \pm 0.1$) matches well with that of nearby

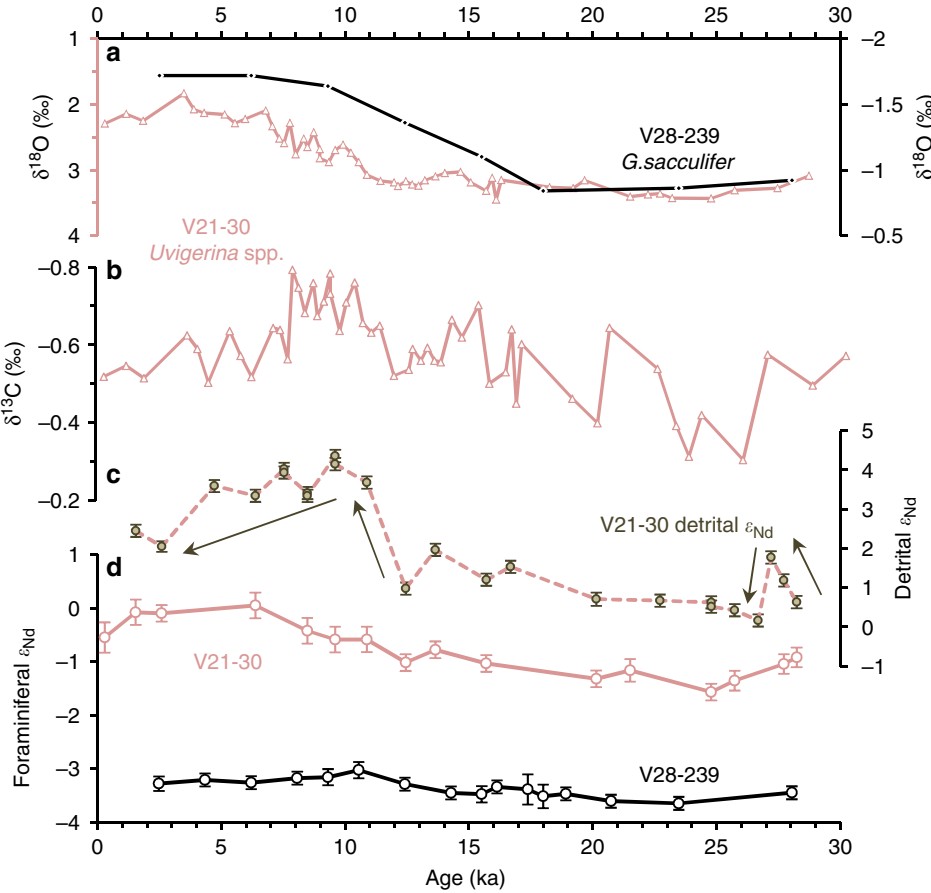

**Fig. 2** Deglacial records of core V21-30 and V28-239 (0–30 ka). **a** Benthic δ¹⁸O of V21-30 (pink)[69] and planktonic δ¹⁸O of V28-239 (black)[53]. **b** Benthic δ¹³C of V21-30 (pink)[69]. **c** Detrital $\varepsilon_{Nd}$ record of V21-30 with arrows denoting changing trends over time. **d** Foraminiferal $\varepsilon_{Nd}$ records of V21-30 and V28-239. Error bars correspond to the 2σ external error of the $\varepsilon_{Nd}$ measurements

seawater value (−3.4 ± 0.4)[24], while those of V21-30 (−0.6 ± 0.3) and ODP1241 (+0.4 ± 0.2) are slightly more radiogenic than ambient seawater signatures (Supplementary Table 10), probably associated with the potential dissolution of detrital fractions after deposition in the EEP[15]. In this case, a quantitative examination of the potential detrital contamination to porewater Nd isotopic signatures over time (see details in the Methods) is beneficial for reliable interpretation of the variation in foraminiferal $\varepsilon_{Nd}$ records.

Our calculation shows the modelled $\varepsilon_{Nd-pw}$ G–I change from detrital contribution can explain only one third of the actual measured foraminiferal $\varepsilon_{Nd}$ G–I change in our parameter setting (Supplementary Figs. 1, 2), which is barely analytically significant. The magnitude, timing, and direction of change in the foraminiferal $\varepsilon_{Nd}$ records still cannot be produced in the $\varepsilon_{Nd-pw}$ records even when we force the detrital contamination to match the core-top and LGM foraminiferal $\varepsilon_{Nd}$ values (Supplementary Fig. 3). Further support for a global rather than local control on our foraminiferal $\varepsilon_{Nd}$ records comes from the resemblance of our foraminiferal $\varepsilon_{Nd}$ records to benthic δ¹⁸O (Figs. 2, 3), which is a proxy for deep water temperature and global ice volume. This would not be the case if the $\varepsilon_{Nd}$ records were controlled by local sediment input. We thus suggest that the foraminiferal $\varepsilon_{Nd}$ variability in our records mainly reflect evolution of bottom water Nd isotopes.

Recent review of seawater and archive $\varepsilon_{Nd}$ values exhibits gradual latitudinal trends in the Atlantic and Pacific at depths

below 600 m, corroborating the effectiveness of Nd isotopes to distinguish between northern/southern sourced water contributions in the intermediate and deep waters[12]. The comparison of Pacific seawater $\varepsilon_{Nd}$ with the conservative tracer like salinity clearly demonstrates the presence of an external Nd source to the Pacific which has a radiogenic composition (Fig. 4), consistent with models of the Nd oceanic cycle[25,26]. The Nd isotopic compositions of deepwater below 3 km are the result of adding the external radiogenic Nd along the LCDW advection over time. Clear local influence is limited in deep waters compared with waters above 3 km. The increased scatter observed in $\varepsilon_{Nd}$-salinity crossplot when moving from deep towards shallower waters is likely caused by the addition of external Nd dominantly from the upper ocean/continental margins rather than from the bottom[27,28]. Another line of evidence comes from the observation of consistent $\varepsilon_{Nd}$ evolution patterns and similar $\varepsilon_{Nd}$ values between adjacent sites in our study. For example, the foraminiferal $\varepsilon_{Nd}$ record of SO136-38 is almost identical to that of Y9 (Fig. 5b, 6b) despite the geographic distance of >260 km between the two sites, because both are recording the SO AAIW at ~1.3 km depth[29]. Likewise, the foraminiferal $\varepsilon_{Nd}$ compositions of RC13-114 closely match those of ODP846 during the last 160 ka (Fig. 3d) although they are more than 1400 km apart in distance and have different $\varepsilon_{Nd}$ in surrounding detrital components[18], because both cores are bathed in NPDW at ~3.3–3.4 km depth in the EEP. Rather than showing 'exceptional disparities between adjacent sites'[28], as would be expected from localized

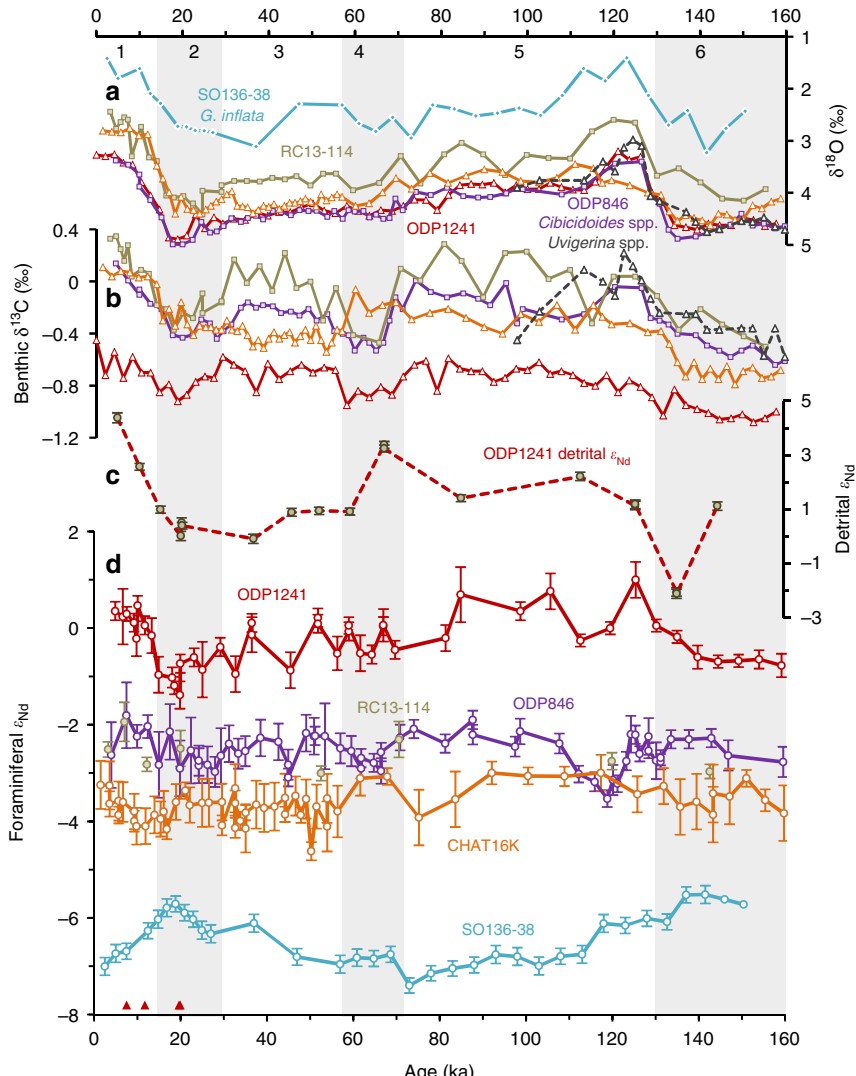

**Fig. 3** Records spanning the last glacial cycle (0–160 ka). **a** Benthic $\delta^{18}O$ and **b** $\delta^{13}C$ records of ODP1241 (red)[55], ODP846 (purple squares for *Cibicidoides* spp. and grey triangles for *Uvigerina* spp.)[70], RC13-114 (brown)[71], CHAT16K (orange)[9] and SO136-38 (cyan)[51]. **c** Detrital $\varepsilon_{Nd}$ record of ODP1241. **d** Foraminiferal $\varepsilon_{Nd}$ records. The red triangles beside the bottom axis represent 4 calibrated AMS $^{14}C$ dates of planktonic foraminiferal species *N. dutertrei* (Supplementary Table 9). Numbers on the top show MIS stages, and the grey bars highlight the glacial stages (MIS 2, 4, and 6). Error bars show the $2\sigma$ external error of the $\varepsilon_{Nd}$ measurements

sedimentary inputs, these spatial and temporal data argue against a benthic input control on the Nd isotopic compositions of the deep ocean and paleo-records[27,28]. These records show that, despite external sources of Nd to the Pacific, the deep water mass hydrography plays the most important role in regulating the spatial $\varepsilon_{Nd}$ pattern and changes through time.

All the currently available foraminiferal Nd isotope records in the open Pacific which cover 0–30 ka and 0–160 ka are compiled in Fig. 5, 6, respectively, and unfortunately there are no such records in the open North Pacific. Except CHAT10K and CHAT16K on the northern Chatham Rise, it is clear that all the South Pacific $\varepsilon_{Nd}$ records show more radiogenic $\varepsilon_{Nd}$ during glacial periods (Fig. 5b, 6b), suggesting reduced contributions of North Atlantic Component Water (NACW)[30]. Moreover, the $\varepsilon_{Nd}$ values converged vertically at the G–I transitions (MIS 6/5 and MIS 2/1) in the South Pacific implying a breakdown of ocean stratification[6,7] and a larger contribution of shoaled glacial NACW[29] or Ross Sea bottom water (RSBW)[7] to the upper Pacific ocean. Since the glacial LCDW and SO AAIW inflow have higher

$\varepsilon_{Nd}$ values than interglacials, a more radiogenic $\varepsilon_{Nd}$ signature during glacial periods would be expected for the North Pacific if the glacial ocean had a circulation regime similar to the modern one[9,11]. However, unlike Y9 and SO136-38, the intermediate-depth core CHAT16K situated on the northern Chatham Rise saw a relatively smooth $\varepsilon_{Nd}$ pattern over the past 160 ka, likely reflecting the $\varepsilon_{Nd}$ signatures of Tasman AAIW which circulates around the South Pacific gyre and mixes its $\varepsilon_{Nd}$ signatures with recirculating southward NPDW and possible Indonesian inputs, although a modification due to local exchange with materials from North Islands could not be excluded[30]. Similarly, the deeper sediment core CHAT10K located close to the North Island and outside the main flow path of the deep western boundary current probably reflects modification by local exchanges[30]. Therefore, it seems the cores located on the main south-to-north flowpath (i.e. in the 'S-Pac' domain of Fig. 4) in the South Pacific exhibit a similar $\varepsilon_{Nd}$ evolution pattern to the Atlantic or Southern Ocean cores as a result of changing propagation of NACW, with a possible influence of RSBW.

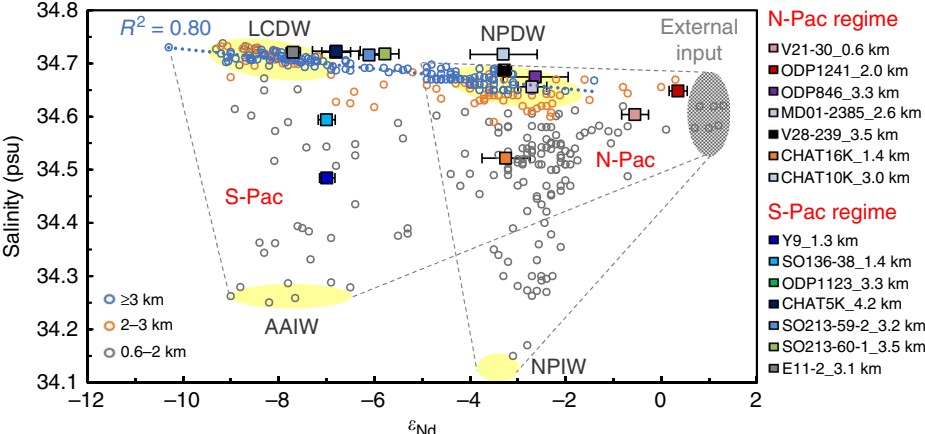

**Fig. 4** Nd isotopic data for seawater and core-top foraminifera plotted against salinity in the Pacific Ocean (>600 m). Salinity data are estimated from World Ocean Altas 2009 using location and depth of seawater $\varepsilon_{Nd}$ measurements, and seawater $\varepsilon_{Nd}$ data are compiled from publications (ref.[14] and references within) and refs.[16,22,24,27,72–82] until March 2018, excluding the unfiltered seawater of which the measured Nd isotope compositions are thought to be contaminated by particulates[83]. The yellow ovals show the characteristic $\varepsilon_{Nd}$-salinity range of the water masses[12]. Note that NPDW is not formed from the sinking of surface water, instead its chemical property is acquired through the ageing of LCDW. The blue dotted trend line ($n = 136$, $R^2 = 0.8$) exhibits a strong linear relationship between $\varepsilon_{Nd}$ and salinity for deep water below 3 km and points to the presence of an external source with high $\varepsilon_{Nd}$ compositions (grey oval). The core-top $\varepsilon_{Nd}$ (see legend for sediment cores with same colour coding in this paper and the data sources in Supplementary Table 1) are plotted against ambient seawater salinity. The scatter of the data points illustrates mixing between different water masses along with additional external input of radiogenic Nd isotopes, which fall into two regions labelled by 'S-Pac' and 'N-Pac'. Except CHAT10K, all the other cores deeper than 3 km reside nicely on the blue dotted ageing line, providing the basis for linking the spatial $\varepsilon_{Nd}$ gradients with deep water transit time. Error bars show the $2\sigma$ external error of the $\varepsilon_{Nd}$ measurements

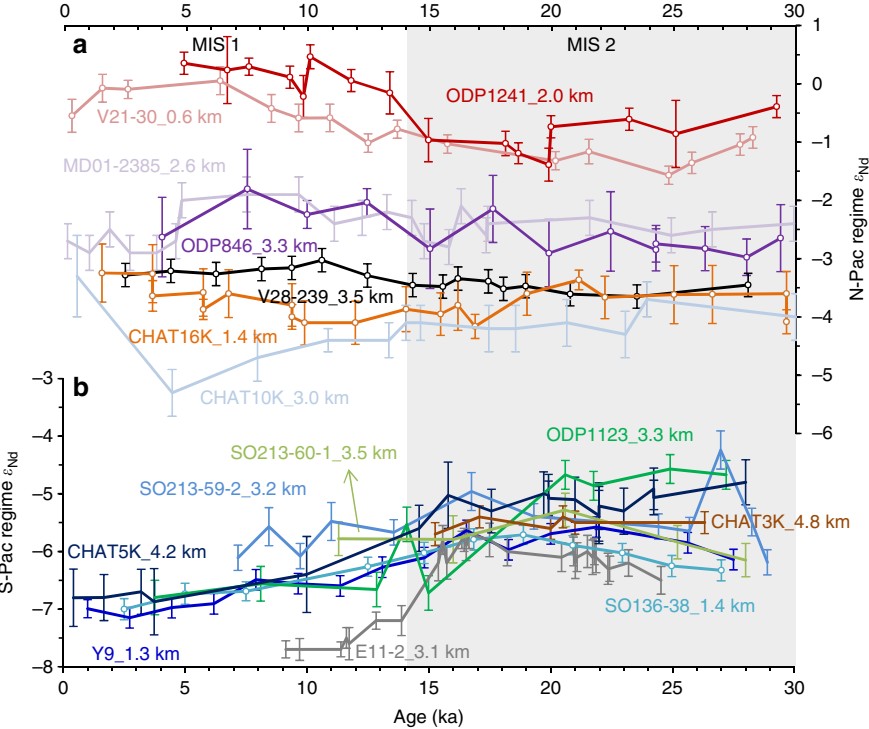

**Fig. 5** A compilation of Pacific foraminiferal $\varepsilon_{Nd}$ records (0–30 ka). **a** N-Pac regime $\varepsilon_{Nd}$ patterns. **b** S-Pac regime $\varepsilon_{Nd}$ patterns. The records with open circle symbols are measured in this study. Data sources of the other records see Supplementary Table 1. The grey bar highlights the glacial stage MIS 2. Error bars show the $2\sigma$ external error of the $\varepsilon_{Nd}$ measurements

In contrast, the higher than modern glacial $\varepsilon_{Nd}$ patterns of LCDW/AAIW in the South Pacific are not transferred downstream to the North Pacific[29]. Instead, the cores there, along with those in the South Pacific but at shallower depths away from

intense northward-flowing currents (such as CHAT10K and CHAT16K), show similar or even less radiogenic glacial $\varepsilon_{Nd}$ signatures (Fig. 5a, 6a). This is not likely to be caused by the input of Nd from increased glacial dust with unradiogenic $\varepsilon_{Nd}$, because

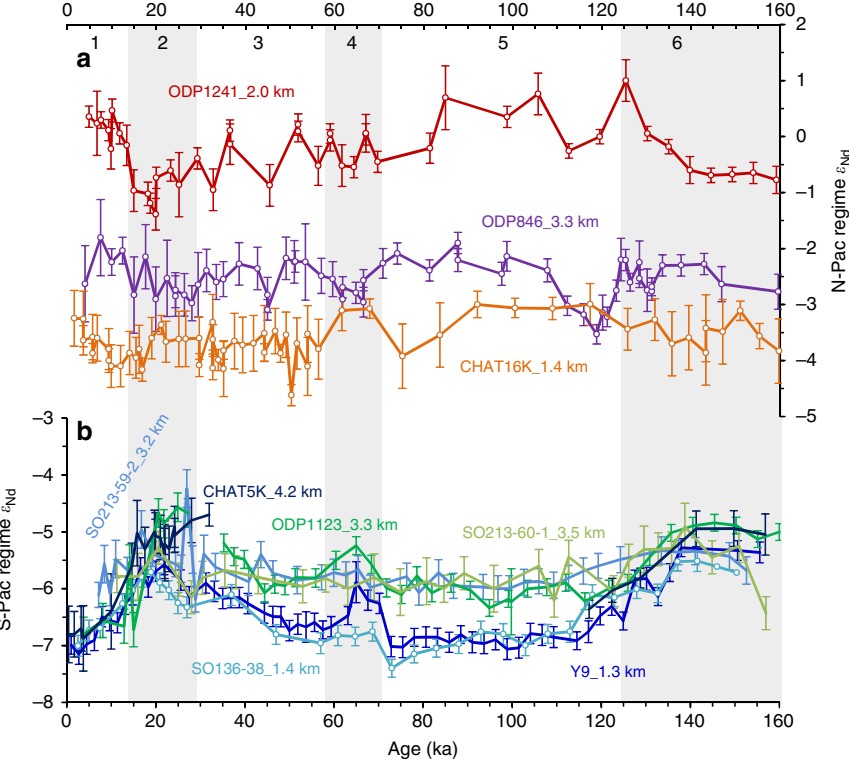

**Fig. 6** A compilation of Pacific foraminiferal $\varepsilon_{Nd}$ records (0–160 ka). **a** N-Pac regime $\varepsilon_{Nd}$ patterns. **b** S-Pac regime $\varepsilon_{Nd}$ patterns. The grey bars highlight the glacial stages (MIS 2, 4, and 6). Error bars show the $2\sigma$ external error of the $\varepsilon_{Nd}$ measurements

Nd budget models suggest that a 2 to 3-fold increase in the Northern Hemisphere dust loads during the ice ages[21] would not affect seawater $\varepsilon_{Nd}$ in the deep Pacific Ocean[25,31]. Based on distinct evolution patterns, the Pacific $\varepsilon_{Nd}$ records can hereby be classified into two types: 'S-Pac regime' and 'N-Pac regime'. Shallower cores in the 'N-Pac regime' (such as V21-30 and ODP1241) are prone to local modification resulting in more radiogenic $\varepsilon_{Nd}$ compositions in the porewater than ambient seawater, while $\varepsilon_{Nd}$ of slowly circulated deepwater is dominantly modified by the accumulation of external Nd input along the flowpath. The spatial $\varepsilon_{Nd}$ gradients below 3 km can therefore be linked with deep water transit time in the Pacific.

The spatial $\varepsilon_{Nd}$ gradients between South and North deep Pacific (ODP1123 and ODP846) decreased from ~4–5 $\varepsilon_{Nd}$ units in MIS 1, 3 and 5 to ~2–3 $\varepsilon_{Nd}$ units during MIS 2, 4 and 6 (Fig. 7c), indicating a reduction in the external Nd input along the advection of LCDW. The diminishment of $\varepsilon_{Nd}$ modification by the radiogenic external Nd on the northward-flowing deepwater has homogenised the spatial Nd isotope signatures during the glacial times. Weathering of continental margins and volcanic islands/arcs in the 'Ring of Fire' surrounding the Pacific Ocean supplies the external source of radiogenic Nd to the deep water through boundary exchange[32] or particle dissolution[33]. Changes in its weathering rate through time are thought to affect the external Nd flux into the ocean. During glacial periods, chemical weathering rate of oceanic islands under colder conditions (assuming average $1.7 \pm 1$ °C cooling for the global surface ocean during the LGM[34]) would be expected to be slightly decreased (~4–15%)[35], while this could be partly balanced by small increase in cation weathering flux by enhanced exposure of island area available for weathering[36]. The counteracting effect thus likely results in negligible G–I variation in continental chemical weathering rates[36,37]. Rather, our data are more consistent with faster Pacific overturning circulation reducing the spatial $\varepsilon_{Nd}$ gradients during the glacial periods.

To carry out a quantitative estimation of changes in the deep Pacific transit time in the past, a box model is developed here (see details in the Methods). Our study provides a first crude estimation for the deep Pacific transit time, which varied between 350 and 1200 years over the last 160 ka (Fig. 7d). The averaged deepwater transit time at late Holocene (4–5 ka) is ~$950 \pm 130$ years, in agreement with the modern tracer observation[38,39]. The deep Pacific transit time in glacial times (MIS 2,4 and 6) was systematically shorter compared to interglacials (MIS 1, 3 and 5), with fastest circulation during peak glacial periods. Deep water $\delta^{13}C$ has been traditionally regarded as a ventilation tracer, and the spatial gradients have shed light on water mass ageing. When LCDW travels across the Pacific, it progressively accumulates nutrients and depleted $\delta^{13}C$ through the remineralization of sinking organic matter, as well as mixing with the overlying waters. The similarity between the $\Delta\delta^{13}C_{846-1123}$ and our $\Delta\varepsilon_{Nd846-1123}$ records (Fig. 7) lends support for our interpretation on the deepwater circulation rates in the Pacific. This interpretation is also supported by sortable silt evidence of ODP1123 in the SWP[8], but interestingly conflicts with bulk leachate $\varepsilon_{Nd}$ records located along formerly glaciated active volcanic margins in the Gulf of Alaska[10], a geological environment where local detrital control has been substantiated[28,40]. Although there are uncertainties in our simplified model, sensitivity tests show that the pattern of faster Pacific overturning circulation during glacial stages is still maintained over the last 160 ka (Supplementary Fig. 4).

The hypothetical Pacific overturning circulation states during the interglacial and glacial stages are synthesized in Fig. 8.

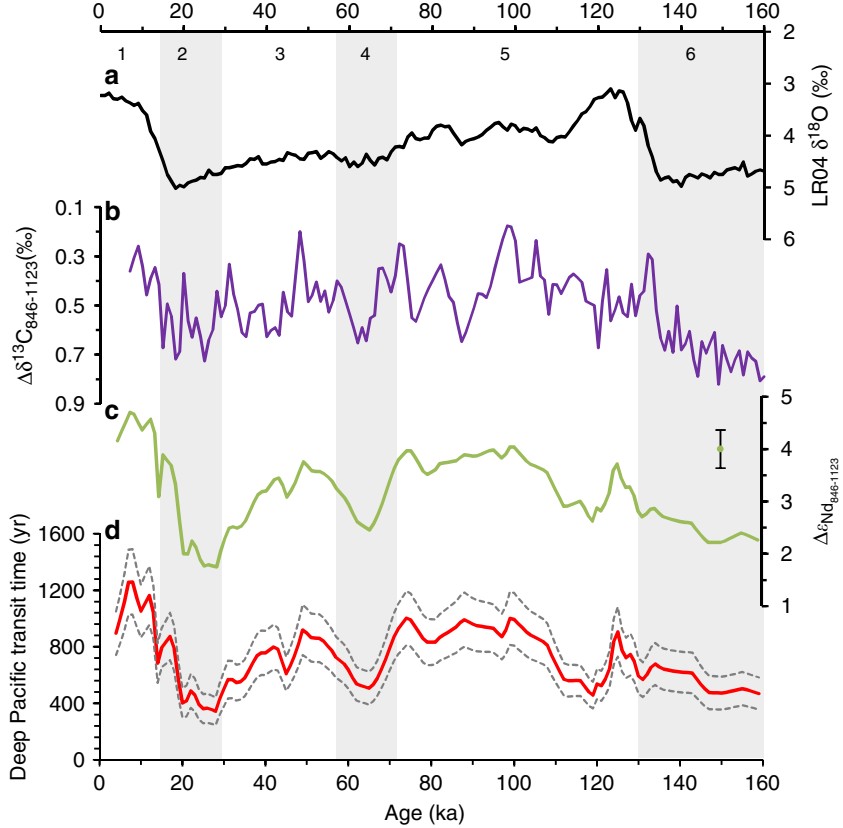

**Fig. 7** Deep Pacific transit time over the last 160 ka. **a** Benthic $\delta^{18}O$ stack[50]. **b** Benthic $\delta^{13}C$ gradients between ODP846 and ODP1123. Note that the benthic $\delta^{13}C$ of ODP846 (*C. wuellerstorfi*)[70] and ODP1123 (*Uvigerina* spp.)[8] are measured on different species and vital effects might affect their $\delta^{13}C$ signatures. **c** Foraminiferal $\varepsilon_{Nd}$ gradients between ODP846 and ODP1123. **d** The deep Pacific transit time estimated from the box model (see parameters in Supplementary Table 10). The grey dash lines display uncertainties propagated from $\varepsilon_{Nd}$ measurements (averaged $2\sigma$ for ODP1123 and ODP846 are 0.19 and 0.31, respectively) into the transit time. The grey bars highlight the glacial stages (MIS 2, 4 and 6)

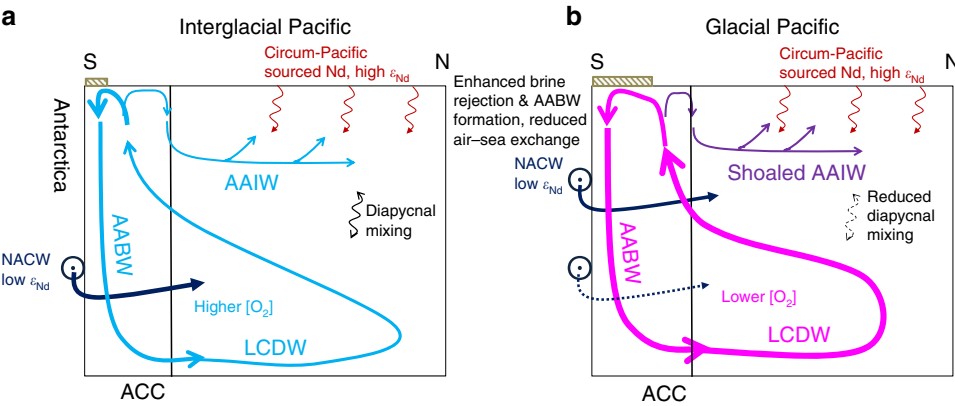

**Fig. 8** The Pacific overturning schematics during the G–I states. **a** Interglacial sketch. **b** Glacial sketch. The vertical black line bounds the northernmost extension of the ACC. Line colours from dark blue, light blue to purple, pink and red indicate increasing $\varepsilon_{Nd}$ values: dark blue North Atlantic Component Water (NACW) with low $\varepsilon_{Nd}$ values, red circum-Pacific Nd sources with radiogenic $\varepsilon_{Nd}$ values. Line thickness changes among panels denote changes in flow flux/rate. The red wavy arrows show the addition of circum-Pacific Nd sources with radiogenic $\varepsilon_{Nd}$ values from the upper ocean to the deep ocean. The black wavy arrows show the diapycnal mixing between different depths. Today, the Atlantic component water with low $\varepsilon_{Nd}$ compositions is propagated into the deep Southern Ocean and incorporated into the LCDW via the ACC. During the glacial periods, with the build-up of sea-ice and enhanced brine rejection, the formation rate of AABW may have increased

Compared with the interglacial state, a faster Pacific overturning circulation is inferred during glacial times, probably driven by enhanced formation and export of AABW[8,41] and/or a northward expansion of RSBW[7] into the Pacific due to increased sea-ice production and brine rejection[42,43]. The Nd isotopes from S-Pac regimes also support a reduced water mass mixing between

LCDW and AAIW during glacials[29], likely associated with reduced diapycnal mixing in a more stratified ocean[44]. The shoaling of NADW in the North Atlantic would lead to more NACW restricted to the upper cell which made their way to the Pacific basin via the eastward flowing ACC. Based on benthic carbon isotopes and radiocarbon evidence, the water mass geometry in the glacial South Pacific was not greatly different to the modern state[6,9]. The water mass boundary between the LCDW and UCDW/NPDW remained at ~3.5 km, while the lower boundary of AAIW may have been slightly shoaled to 1.2 km[45]. The potential increased carbon stock in the deep Pacific inferred from low oxygenated deep water[2,46] would probably not be related to reduced circulation strength, but rather indicating lower oxygen concentration in the Southern Ocean[46,47], in accordance with depleted radiocarbon observations[3,48]. We conclude that a faster but less ventilated overturning circulation might have operated in the glacial Pacific Ocean, and mechanisms such as increased biological pump efficiency and poor high latitude air-sea exchange[49] may be responsible for atmospheric $CO_2$ drawdown in the deep Pacific Ocean during glacial periods. Our results here thus put constraints on ocean circulation models to better simulate the atmospheric $CO_2$ change through past G–I cycles.

## Methods

**Sediment cores**. Foraminiferal Nd isotopes are measured on seven new cores in this study, including two from the SWP, one from the WEP and four from the EEP.

CHAT16K and SO136-38 (at ~1.4 km water depth) are located to the north and south of the Chatham Rise, recording the signatures of two different AAIW types, i.e. Tasman AAIW and SO AAIW (also termed as $AAIW_N$ and $AAIW_S$[9]), respectively (Fig. 1a). The age model of the northern site CHAT16K is slightly modified by retuning the *Uvigerina peregrina* $\delta^{18}O$ record[9] to LR04 stack[50]. New $\varepsilon_{Nd}$ measurements on CHAT16K is performed, extending the existing deglacial record[30] to 160 ka. The age model of the southern site SO136-38 is constructed by correlating $\delta^{18}O$ profile of *Globorotalia inflata*[51] to a new global planktonic $\delta^{18}O$ stack[52].

The western equatorial core V28-239 was raised from the Solomon Plateau at a water depth of 3490 m. The age model is developed by tuning *Globigerinoides sacculifer* $\delta^{18}O$ record[53] to the planktonic $\delta^{18}O$ stack[52].

Four sediment cores in the EEP were taken from water depth between 617 and 3436 m, with foraminiferal Nd isotopes analysed on all the cores and detrital Nd isotopes on two sites (ODP1241 and V21-30). The shallowest core V21-30 is situated in the modern oxygen minimum zone, with an average sedimentation rate of 13 cm ka$^{-1}$ over the past 30 ka based on radiocarbon age[54]. The age model of ODP1241[55] is modified by measuring 4 new $^{14}C$ dates (*Neogloboquadrina dutertrei* > 300 μm) for the upper 60 cm of Hole B (Supplementary Table 9). Deeper cores ODP846 and RC13-114 are bathed at similar water depths (~3.3–3.4 km), and their foraminiferal $\varepsilon_{Nd}$ compositions are compared to check the reliability of this archive in recording water mass $\varepsilon_{Nd}$ signatures.

**Nd isotope measurements**. Nd isotopes are measured on both foraminifera (for all the seven cores) and detrital fraction of the bulk sediments (for EEP cores V21-30 and ODP 1241). The former is used to trace the porewater Nd isotope changes, whereas the latter is used to identify changes in provenance and inputs of lithogenic materials.

Foraminifera for each sample were prepared following a routine protocol[13]. In brief, 30–100 mg planktonic foraminifera of each sample were handpicked from >63 μm fraction. All clays were removed by ultrasonication after the foraminiferal tests were broken open. The physically cleaned foraminifera were then dissolved in 1 M acetic acid.

For detrital analysis, 3–5 g sediment was repeatedly decarbonated overnight using a buffered acetic acid solution until the lack of $CO_2$ production indicating no carbonate remained. Then the decarbonated samples were MilliQ water rinsed and leached twice with a 0.02 M solution of hydroxylamine hydrochloride in 25% (v/v) acetic acid. The residues were then MilliQ rinsed, dried and soaked in 10% peroxide to remove organic matter. After drying down again, the detrital samples were sequentially digested by the mixture of concentrated $HNO_3$ and HF for at least 48 h at 110 °C and by aqua regia for another 24 h at 80 °C.

After dissolution, all samples underwent a two-step ion chromatographic separation (Eichrom TRUspec and Eichrom LNspec resin) to isolate and purify Nd. The isotopic ratios of Nd were analysed on a Nu Plasma or Thermo Neptune Plus multi-collector inductively-coupled plasma mass spectrometer (MC-ICP-MS) in the Department of Earth Sciences at the University of Cambridge. Procedural

blanks for foraminiferal and detrital samples were negligible (less than 0.3% and 0.07‰, respectively) and were therefore not corrected for. Instrumental mass bias was corrected by applying an exponential mass fractionation law using $^{146}Nd/^{144}Nd$ of 0.7219 and each sample was bracketed with the concentration-matched reference standard JNdi-1, the measured composition of which was corrected to the accepted value of $^{143}Nd/^{144}Nd = 0.512115$. The external reproducibilities ($2\sigma$) of Nd isotope measurement during each analytical session are given by repeated measurements on concentration-matched JNdi-1 standards. The average $2\sigma$ external reproducibility for 20 ng of Nd is ~0.2 (on Neptune Plus) and ~0.5 (on Nu Plasma) $\varepsilon_{Nd}$ units, while larger errors for some samples reflect low Nd abundance due to small sample sizes.

**Examination of detrital influence on porewater $\varepsilon_{Nd}$ records**. The influence of detrital dissolution to porewater $\varepsilon_{Nd}$ for V21-30 and ODP1241 over the studied periods is estimated via a binary mixing model: the $\varepsilon_{Nd}$ of the sedimentary pore-water ($\varepsilon_{Nd-pw}$) is determined by a mixing of seawater derived Nd (authigenic) and lithogenic particulate Nd (old continental and young volcanic arc material). The relationship is expressed as follows:

$$\varepsilon_{Nd-pw} = \frac{\varepsilon_{Nd-au} \times F_{au} + \varepsilon_{Nd-vol} \times \alpha_{vol} F_{vol} + \varepsilon_{Nd-cont} \times \alpha_{cont} F_{cont}}{F_{au} + \alpha_{vol} F_{vol} + \alpha_{cont} F_{cont}} \quad (1)$$

All the parameters and their values used in this model are shown in Supplementary Table 10. In fact, modern oceanic Nd isotope budget has been constrained in many previous studies[25,56], which allows us to estimate the authigenic Nd flux ($F_{au}$) to the sea floor. A mean authigenic Nd flux in the Pacific Ocean ($F_{mean}$) is calculated based on Nd residence time ($\tau = $ ~500 yr)[25] as follows:

$$\tau = \frac{C_d \times \rho V \times M_{Nd}}{F_{mean} \times A} \quad (2)$$

Using the parameters in Supplementary Table 10, we obtain a mean authigenic Nd flux ($F_{mean}$) in the Pacific of ~$2.76 \times 10^{-5}$ g m$^{-2}$ yr$^{-1}$. This number is in agreement with previous INDOPAC box estimation using 10-box model[25]. Settling particles are considered to be the main flux of particulate material to the seafloor. The authigenic Nd flux thus varies with different particle fluxes, resulting in higher flux in the marginal region and much lower flux in the pelagic central Pacific region[56]. The adopted $F_{mean}$ should be taken as the lower limit of authigenic flux for our EEP cores and thus is a conservative estimation.

Given a rich carbonate (~60–80%) and low diatom contents (<10%) of V21-30 and ODP1241 in our studied intervals[17,57], the lithogenic Nd flux ($F_{litho}$) composed is taken as the non-carbonate sedimentary flux when a detrital Nd concentration ($C_p$) is applied:

$$F_{litho} = DBD \times LSR \times (1 - X_{CaCO3}) \times C_p \quad (3)$$

Linear sediment rate (LSR) is calculated based on the age model. Constant dry bulk density (DBD = 0.63 g cm$^{-3}$), low-resolution $CaCO_3$% data points of ODP1241 are from the initial IODP report (~60–70%)[17], and the rest $CaCO_3$% are extrapolated based on exponential relationship with measured coarse fraction%[55]. The shallower core V21-30 has high $CaCO_3$%, and the contents of MIS 1 and MIS 2 are represented by the measurements of core-top = 70%[58] and LGM = 77%[59], respectively. The core-top and LGM Nd concentrations for the detritus of ODP1241 ($C_p$) we measured are very similar (15.0 and 14.8 ppm, respectively), and thus a constant value of 15 ppm is taken for both cores for simplicity.

The lithogenic detrital Nd flux is composed of continental flux ($F_{cont}$) and volcanic flux ($F_{vol}$):

$$F_{litho} = F_{vol} + F_{cont} \quad (4)$$

The proportions of old continental particles ($X_{cont}$) and volcanic (1-$X_{cont}$) components over time can be further calculated based on a two-endmember mixing model: the detrital $\varepsilon_{Nd}$ is determined by a mixing of Nd from old continental particles (average $\varepsilon_{Nd-cont} = -10$)[23] and Central America/Galapagos volcanic arcs (average $\varepsilon_{Nd-vol} = +7$)[22] as follows:

$$\varepsilon_{Nd-d} = \varepsilon_{Nd-cont} \times X_{cont} + \varepsilon_{Nd-vol} \times (1 - X_{cont}) \quad (5)$$

$$F_{cont} = F_{litho} \times X_{cont} \quad (6)$$

The Nd dissolution factors ($\alpha$) of the corresponding components are assumed to be constant with respect to time[25]. The proportion of soluble Nd in marine particles is thought to be highly variable[60–63], but literature suggests that the basaltic material is usually more reactive compared to continental/granitic material[64] and has been replicated in closed-system experiments[65]. $\alpha_{cont} = 2\%$

derived from laboratory experiment[60] is commonly used in Nd budget calculations[26,56,66], while there is no common $\alpha_{vol}$ so far in the literature, ranged from < 0.4%[67] to ~1.5–8.5%[68] based on batch reactor experiments.

We have taken approach that first making the core-top $\varepsilon_{Nd-pw}$ value match the measured foraminiferal $\varepsilon_{Nd}$ in Eq. (1). Assuming a constant authigenic Nd flux ($F_{au} = F_{mean} = 2.76 \times 10^{-5}$ g m$^{-2}$ yr$^{-1}$), we obtain $\alpha_{arc} = 3\%$ for V21-30 and $\alpha_{arc} = 9\%$ for ODP1241, both of which are within the range of reported marine particle dissolution factors. This means the discrepancy between the seawater and core-top foraminiferal $\varepsilon_{Nd}$ for our EEP cores could be explained by detrital contamination of 2% continental particle dissolution and 3–9% volcanic material dissolution during diagenesis after burial. The parameters are then applied back in time to derive the $\varepsilon_{Nd-pw}$ records of both cores in Figs. 1c, 2c.

Considering the uncertainties of the parameters such as the true authigenic Nd flux ($F_{au}$) and the wide range of dissolution factors of volcanic and continental materials ($\alpha_{vol}$ and $\alpha_{cont}$), a sensitivity test is performed by creating the same G–I change in porewater $\varepsilon_{Nd}$ records artificially to see how the $\varepsilon_{Nd-pw}$ calculated compared to our measured foraminiferal $\varepsilon_{Nd}$ through time (Supplementary Fig. 3).

**Box model for estimating deep Pacific transit time.** In this model, the deep Pacific Ocean is treated as a box with LCDW exclusively ventilated from the Southern Ocean, and the water mass $\varepsilon_{Nd}$ signatures are assumed to be gradually modified in the process of northward LCDW advection by time-dependant accumulation of external Nd input from a constant source[15]. To carry out such work the basic principle for selecting the tracer is that it could have a residence time comparable to the oceanic transit time, i.e. a few hundred years. In the upper, ocean Nd isotopes is actively recycled by particles[62,63], which will lead to short residence time. In this regard, ODP1123 and ODP846 are chosen to calculate the deep transit time from the northward-flowing LCDW in the SWP to the southerly return flow in the EEP as regional/local influence on Nd isotope compositions for deep water below 3 km and away from continental margins should be limited[28,33]. Assuming steady state, the scavenged Nd has the same isotope composition as in-situ seawater and the concentration of Nd ($Q_{Nd}$) equals to the sum of Nd brought by the LCDW inflow ($Q_{LCDW}$, which is supposed to be constant over time) and the accumulation of external input over transit time ($t$). The reference flux of external input ($f_{EI} = 3.24 \times 10^9$ g yr$^{-1}$) is averaged from previous model estimations for the Pacific Ocean[25,66], and the reference $\varepsilon_{Nd}$ value of external input ($\varepsilon_{Nd-EI} = +1$) is adopted from earlier model of oceanic Nd budget[25]. The $\varepsilon_{Nd}$ of the deep water along the pathway over time can thus be calculated by the mass and isotope balance as follows:

$$\varepsilon_{Nd-LCDW} \times Q_{LCDW} + \varepsilon_{Nd-EI} \times f_{EI} \times t = \varepsilon_{Nd} \times (Q_{LCDW} + f_{EI} \times t) \qquad (7)$$

All the parameters and their values used in this model are shown in Supplementary Table 10. The reliability of our calculation based on this equation is supported by the ability to reproduce modern deepwater $\varepsilon_{Nd}$ distribution[12].

Considering the uncertainties and the potential variability of the external Nd input into the deep ocean over time, sensitivity tests have been performed varying $\varepsilon_{Nd-EI} = 0, +0.5, +1, +1.5, +2$ and $f_{EI} = 2.84, 3, 3.24, 3.4, 3.63 \times 10^9$ g yr$^{-1}$, respectively, for reconstructing the deep Pacific transit time during the last 160 ka (Supplementary Fig. 4).

## Data availability
The data reported in this paper are provided in the Supplementary Information.

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

## Acknowledgements

R.H. is supported by Global Change Program of Ministry of Science and Technology of China (2016YFA0600503) and China Postdoctoral Fund (2017M620201) during writing of this paper and the China Scholarship Council for her living cost in Cambridge. The analytical costs were covered by NERC project NE/K005235/1 to A.M.P. We thank the Gulf Coast Repository for providing the sample materials of ODP1241 and ODP846, the Lamont-Doherty Core Repository for providing the sample materials of V21-30 and RC13-114 and R. Anderson for helpful discussion, H. Bostock and H. Neil from the National Institute of Water and Atmospheric Research (New Zealand) to provide the sample materials of SO136-38, J. Clegg, V. Rennie and S. Crowhurst for technical support.

## Author contributions

R.H. and A.M.P. designed the study, R.H. performed the work and R.H. wrote the manuscript with inputs from A.M.P.

## Additional information

**Competing interests:** The authors declare no competing interests.

