## [Peer Review File · Nature Communications]

Reviewer #1 (Remarks to the Author):

Review of Hu & Pietrowski “Neodymium isotope evidence for glacial-interglacial variability of deepwater transit time in the Pacific Ocean“, submitted to Nature Communications

The paper presents a number of new Nd records from the deep Pacific in order to constrain orbital-scale changes in the Pacific overturning circulation across the last glacial-interglacial climate cycle (including isotope stages 1 to 6). The overturning circulation in the Pacific is one of the large unknowns in our knowledge of Quaternary paleoceanographic changes and their relation to potential storage of atmospheric CO₂. As such the new records by Hu & Pietrowski clearly merit publication in a high impact journal such as Nature Communications.

The different geochemical records (seven novel and ten previously published sediment cores) are mainly from the South Pacific and the tropical sector of the Pacific. There are no records from the North Pacific itself (which could be mentioned somewhere in the text). The authors discuss the different records in quite some detail focussing on the main orbital trends, i.e. the glacial-interglacial variations. In this context, they distinguish a S-Pac versus N-Pacific regime. The southern one is characterized by more radiogenic ϵNd during glacials interpreted as reduced North Atlantic component water consistent with previous findings. From this, given the modern circulation strength, a more radiogenic glacial ϵNd signature would be expected for the North Pacific. In contrast, the records indicate similar or less radiogenic values in the tropical Pacific and shallower South Pacific return low (the N-Pacific regime). I think these conclusions are well documented and supported by the data. The only pitfall might be, that records from North Pacific itself are not included in the core collection used by the authors.

The very important next step which makes this paper really suitable for a more general readership, is the transfer of the ϵNd signature changes into quantitative estimates, in this case deep Pacific transit time of water masses over the last 160 kyr. Hu & Pietrowski use a box model in order to estimate these transit times. The model is described in the supplementary material – a detailed evaluation of the suitability of this model is unfortunately beyond my expertise. In any case, the overall results are very important and provide evidence against the commonly assumed reduced Pacific overturning circulation during glacials. Instead, the reduced spatial ϵNd gradients and the box model results indicate substantially enhanced Pacific transect times during glacials, especially MIS2. If true, this finding has important implications for our mechanistic understanding of atmospheric CO₂ drawdown during glacials.

The paper is generally very precisely written. The results are well presented and the discussion is generally fine. One might add some additional records for deep Pacific circulation (e.g. proxies other

than Nd) to a kind of summary figure. Some of those are discussed in the text but are not really shown in any figure (final chapter of the paper).

A minor remark regarding Figure 1:

I would split the zonal section into an eastern and western Pacific meridional section and may be a northern tropical zonal one.

Taken together, I think the extensive and novel Nd records and very challenging interpretation presented in this manuscript merits publication in Nature Communication.

Reviewer #2 (Remarks to the Author):

Hu and Piotrowski's 'Neodymium isotopic evidence for glacial-interglacial variability of deepwater transit time in the Pacific Ocean' is an interesting manuscript where the authors have taken an approach (rather a new approach in my view) to interpret Nd isotopes to derive transit time of deepwater in the Pacific. Their conclusion is interesting in the sense they find the Pacific to be more dynamic during glacial periods compared to the Atlantic. They hypothesized that high latitude Southern Ocean processes were responsible for a faster deep ocean circulation in the Pacific during glacial times. This is an important finding. The overall study is well thought out and the manuscript is well written.

I do have some comments and suggestions that are enumerated below.

Comments:

Working with Nd isotopes in the Pacific has always been a challenge since the water masses in the Pacific are not as distinct (in terms of Nd isotopic signature) as in the Atlantic. On the top of that, Pacific is full of volcanic materials and more often than we would like, these volcanic materials affect the water mass Nd isotopic signature and derail its water mass tracing property. More recently, a series of papers off Oregon have shown that pore water flux could have equally affect the bottom water values.

The main interpretation in this manuscript is contingent on demonstrating that Nd isotopes from Fe-Mn coating off foraminifera are pristine bottom water signal. The authors analyzed contemporaneous sediment samples from the same core as the foraminifera derived records to establish that non-seawater sources did not control the foraminifera derived Nd isotopic values. They have reported detrital values for two cores V21-30 and ODP 1241, both from equatorial Pacific

and argued that patterns of changes in the detrital values are independent of foraminifera derived record. The authors pointed out that the detrital materials did not change in accordance with G-IG cycles and rather speculated the role of moving ITCZ as potential sources of external Nd. The ITCZ is also supposed to move on a G-IG cycle, so I am not sure what exactly the authors wanted to say here? So it is necessary that they clarify this a bit more. Also, I am curious how studies of past ITCZ shifts (Jacobel et al., 2016) from the Pacific compare to their detrital record?

The detrital record of ODP 1241 is problematic specifically the last deglacial part. To me it appears that the detrital and foraminifera derived records follow the general pattern of change during the last deglacial. I would be very careful to claim that this part of the record is contamination free. If the foraminifera derived Nd isotopes following the last deglacial indeed represent contamination from detrital sediment, it becomes increasingly difficult to argue that the older part of the record is pristine seawater. The authors should address this issue in further detail. Moreover, ODP 1241 core top foraminifera Nd isotopic signature is 0.35, quite different from nearby modern seawater value at comparable depth (Grasse et al., 2012; Stn. 160). Wouldn't you expect seawater Nd isotopes to agree with core top foraminifera value?

The statement about porewater not affecting the bottom signal is well intentioned but it needs to be backed up with evidence. At this point it is a statement with some circumstantial evidence. While I am not sure how much of this is possible within the scope of the work presented here, it is worth the time to try a simple box model to show how much contribution from porewater is needed in order to create the downcore seawater Nd isotopic pattern that is observed in their record. If the model derived % contribution from porewater is anomalously high, it might be grounds to disregard the porewater contribution. Parameterization of such a model can be done using the published numbers off Oregon.

The Chatham Rise core CHAT16K does not show a lot of variations for the last 160 ka. The authors claimed that it is due to mixing between Tasman AAIW and southward circulation NPDW. It can also be argued to be modification due to local exchange (given how close it is to the continents and volcanic sources); can the authors strengthen their argument?

Minor comments:

Figure 5b is almost impossible to follow. Too busy.

Figure 6: Needs legends.

In summary, the approach taken by the authors to derive the transit time of deep water in the Pacific is new and quite interesting. While the modeling component is simple, it is a good start and hopefully there will be more systematic modeling attempts in the future.

Reference:

A. W. Jacobel, J. F. McManus, R. F. Anderson, G. Winckler, Large deglacial shifts of the Pacific Intertropical Convergence Zone. *Nat. Commun.* 7, 10449 (2016).

Reviewer #3 (Remarks to the Author):

Hu and Piotrowski use a combination of compiled and new Nd isotope records from foraminifera throughout the Pacific Basin to develop records of seawater Nd isotopic evolution over the past 160 ky. This is augmented by detrital Nd isotope data from one region in the EEP. They use these data in a simplified model to argue for more rapid circulation in the Pacific during glacial times. This result has implications for understanding ocean processes on glacial- interglacial timescales as well as carbon storage in the glacial ocean. There are some interesting aspects of the manuscript that make it appropriate for *Nature Communications*, but there are also aspects that require clarification and I worry it may be a bit of a challenge to address the changes I recommend below in a short journal format.

The authors do a nice job of setting up the importance of their work in the abstract and Introduction, but they could do a better job of summarizing the implications of their results within the framework of that set up. This might be accomplished effectively with a well-constructed Conclusion section. Some of that information is in the last section, but it would be helpful to the reader to wrap up the major implications succinctly. Along those lines, there could also be a comment at the end of the abstract to explain the implication of the statement that the glacial Pacific did not operate in a manner similar to the glacial Atlantic in terms of carbon storage.

The manuscript is organized in a reasonable fashion, but I also think the second author could give the manuscript a good read through to catch several grammatical errors. Without page, section or line numbers it is difficult to refer to specific aspects of the text. For this reason I have added minor comments to the word version of the text (attached) rather than at the end of this review. I also

numbered the pages starting with the introduction and will refer to page numbers that way. Due to the addition of my comments, my page numbers may not match those in the original text.

In order to reach their conclusion about rapid glacial circulation in the Pacific, Hu and Piotrowski need to demonstrate that 1) glacial-interglacial (gl/ig) variations in ϵ_{Nd} of Pacific foraminifera are not driven by variations in detrital inputs and subsequent reactions within the sediment, in other words, the water chemistry is changing, not the detrital inputs and modification, and 2) the cause of the gl/ig isotopic variations in the water is caused by changes in the residence time of the water rather than changes in the ϵ_{Nd} value or flux from detrital sources that react with the waters along the flow path. At some level these two points are very similar, but they have chosen to address point one through a discussion of measured detrital ϵ_{Nd} values from one region, and point #2 through modeling.

One of my concerns relative to point 1 is that the flux of sediment and associated impacts on porewater chemistry and surface exposure time might influence the foraminiferal Nd isotopes. The detrital section (pg 9-10) is an attempt to argue that the detrital inputs are not the cause of the changes observed in water chemistry. They appear to assume that the "equilibrium time" can be equated with the age of the sample, but perhaps it has more to do with time of foram exposure to seawater (sedimentation rate) or reactivity of the sediment. Pure carbonate sediments are likely to have very low porewater [Nd], but the addition of detrital material could significantly increase that concentration, as Abbott et al. (2015) showed. In addition, ash is likely to be more reactive than other continental sources (references by Wilson). In other words, there are clearly changes in seawater ϵ_{Nd} on gl/ig timescales, but there are also likely to be changes in sedimentation. How can they eliminate the sedimentation changes as the source of the ϵ_{Nd} variations?

They could strengthen their argument for a seawater source of Nd in the 'Foraminiferal versus detrital Nd isotope compositions' section substantially by expanding their discussion on dust sources and providing some context for the magnitude of inputs through time. Additional detrital data from a region other than the EEP, which is near the end of the circulation cell, would also be useful, but time consuming to obtain. Given the data they have, it would help to describe potential detrital endmembers more broadly rather than simply focusing on EEP sources, and to be more precise in their discussion of EEP sources, which I found rather confusing. Hu and Piotrowski imply the only detrital source in this region is arc material; however, they also discuss riverine material, but then state that also has an arc composition. I was confused about the distinction. They play with variations in endmember compositions, but don't clearly define the processes leading to those variations. Why wouldn't loess inputs also increase during dustier glacial times? What is the distinction between 'volcanic' and 'riverine' arc inputs?

A more effective evaluation of the detrital contributions might include determining the percent detrital material (which might be equivalent to the inverse of % $CaCO_3$?) and sedimentation rate for

each sample to see if there is a relationship between the foram-detrital Nd isotope offset and the amount of detrital material or exposure time; although, Wilson, from this research group, has pointed out that the composition of the detrital material also affects its reactivity. The general increase in foram ϵ_{Nd} with increasing detrital ϵ_{Nd} during terminations is what catches the eye in Figs. 3 and 4, and if they could demonstrate that the foram response isn't linked through an increase in the rate of detrital inputs or the reactivity of the detrital inputs, it would strengthen their rather weak argument that "the evolution of bottom water Nd isotopes rather than detrital contamination has driven the foraminiferal ϵ_{Nd} variability in our records." I recognize that this is a short format journal, but I think much of this discussion could replace the existing discussion rather than simply building on it.

I also had a difficult time following the discussion for the N versus S regime patterns, starting with the discussion on pg 14. In order to see the patterns they present, it would be very helpful to label the sites and depths for each line (perhaps in the appropriate color) for Figs. 5 and 6 because it is difficult to make out the subtle color differences in the short lines of the legend. It would also be a good idea to make the spacing along the Y axes the same for panels a and b to make comparisons easier. The point about higher ϵ_{Nd} values during glacials (less NACW) than interglacials is easy to see in the plots and makes sense, but I could not follow the discussion about the 'vertical convergence' at gl-ig transitions. I can see that there is less stratification during the glacials (which is most apparent in Fig 7c), but I don't understand the point they are trying to make about the gl-ig transitions, and Fig 7c makes it clear that the transitions are not the time of minimum offset. The next step in the discussion about decreased modification of ϵ_{Nd} during northward flow makes sense as an explanation for the minor reduction in the intermediate to deep offset in the North Pacific seen in glacials, but that decrease is very subtle and seems to vary from glacial to glacial, such that MIS 6 does not appear to reflect the decrease at all. Is there a way to highlight the offset for the reader, such as a plot of the difference between the two records? Initially, I assumed that the changes in the offset between northern and southern regimes was a function of whether or not that southern waters influenced by NACW were actually flowing north, thus I think they could state more clearly that they are arguing LCDW is flowing north throughout the record, but the seawater ϵ_{Nd} is modified more during slower transport during ig times than during the rapid transport during gl times. This statement is definitely in their text but it took a few readings for me to get there.

I do not do a lot of modeling and I'm not sure I exactly followed all of their description of what they did in the model, but it appears to be a very simplified view of the system. I have no objection to this type of simplification as long as the assumptions are clearly stated and justified, and the simplifications lead to robust results with interesting implications. One of my concerns about the short format for this paper is that I don't think the assumptions should be buried in the supplement. It's fine to put the details in the supplement and only include a summary of the results in the figures in the paper (as in Fig.7), but the reader needs enough information to evaluate the interpretations, and that is not the case in the current version of the manuscript. I think it is problematic to have the statement that "This model is a very simplified scheme which does not differentiate various sources of external Nd inputs with different Nd isotope compositions and fluxes since the main purpose for

this model is to estimate the deep Pacific transit time instead of Nd cycling process” in the supplement and then the statement that “A simple mass balance calculation combined with paleoceanographic observations suggest the seawater ϵNd cannot be explained by changes in the external Nd input alone” within the text. It seems that the first statement is telling me that they can’t confidently evaluate the point they make in the second statement. Is it more accurate to simply say that the model illustrates that more rapid overturning in the glacial could produce their observations? Can they provide more information about the assumptions and outcomes of the model with reference to variations in detrital inputs to justify their statement that the sediment modification of the seawater ϵNd alone could not produce the observed pattern? In order to do that, I think they need to explain more clearly how the sediment might vary over gl/ig, and preferably back that up with some data.

In summary, I think Hu and Piotrowski have reached an interesting conclusion about variations in circulation in the Pacific during g/ig that has important implications for understanding the carbon cycle and impacts of climate change. Thus, their results are appropriate for Nature Communications; however, they need to clarify a few points related to the potential for changes in sedimentation on gl/ig timescales to drive the seawater variations as noted above and provide more of an explanation of the model, particularly the assumptions that go into the simplification of the system, in the main text. If they can do that within the constraints of the paper limit, I would support publication in Nature Communications. Note, specific comments on wording and figures are included in track changes in the attachment.

Reviewers' comments:

Reviewer #1 (Remarks to the Author):

Review of Hu & Piotrowski "Neodymium isotope evidence for glacial-interglacial variability of deepwater transit time in the Pacific Ocean", submitted to Nature Communications.

The paper presents a number of new Nd records from the deep Pacific in order to constrain orbital-scale changes in the Pacific overturning circulation across the last glacial-interglacial climate cycle (including isotope stages 1 to 6). The overturning circulation in the Pacific is one of the large unknowns in our knowledge of Quaternary paleoceanographic changes and their relation to potential storage of atmospheric CO₂. As such the new records by Hu & Piotrowski clearly merit publication in a high impact journal such as Nature Communications.

The different geochemical records (seven novel and ten previously published sediment cores) are mainly from the South Pacific and the tropical sector of the Pacific. There are no records from the North Pacific itself (which could be mentioned somewhere in the text). The authors discuss the different records in quite some detail focussing on the main orbital trends, i.e. the glacial-interglacial variations. In this context, they distinguish a S-Pac versus N-Pacific regime. The southern one is characterized by more radiogenic ϵNd during glacials interpreted as reduced North Atlantic component water consistent with previous findings. From this, given the modern circulation strength, a more radiogenic glacial ϵNd signature would be expected for the North Pacific. In contrast, the records indicate similar or less radiogenic values in the tropical Pacific and shallower South Pacific return flow (the N-Pacific regime). I think these conclusions are well documented and supported by the data. The only pitfall might be, that records from North Pacific itself are not included in the core collection used by the authors.

The very important next step which makes this paper really suitable for a more general readership, is the transfer of the ϵNd signature changes into quantitative estimates, in this case deep Pacific transit time of water masses over the last 160 kyr. Hu & Piotrowski use a box model in order to estimate these transit times. The model is described in the supplementary material – a detailed evaluation of the suitability of this model is unfortunately beyond my expertise. In any case, the overall results are very important and provide evidence against the commonly assumed reduced Pacific overturning circulation during glacials. Instead, the reduced spatial ϵNd gradients and the box model results indicate substantially enhanced Pacific transect times during glacials, especially MIS2. If true, this finding has important implications for our mechanistic understanding of atmospheric CO₂ drawdown during glacials.

The paper is generally very precisely written. The results are well presented and the discussion is generally fine. One might add some additional records for

deep Pacific circulation (e.g. proxies other than Nd) to a kind of summary figure. Some of those are discussed in the text but are not really shown in any figure (final chapter of the paper).

A minor remark regarding Figure 1:

I would split the zonal section into an eastern and western Pacific meridional section and may be a northern tropical zonal one.

Taken together, I think the extensive and novel Nd records and very challenging interpretation presented in this manuscript merits publication in Nature Communication.

Reviewer #2 (Remarks to the Author):

Hu and Piotrowski's 'Neodymium isotopic evidence for glacial-interglacial variability of deepwater transit time in the Pacific Ocean' is an interesting manuscript where the authors have taken an approach (rather a new approach in my view) to interpret Nd isotopes to derive transit time of deepwater in the Pacific. Their conclusion is interesting in the sense they find the Pacific to be more dynamic during glacial periods compared to the Atlantic. They hypothesized that high latitude Southern Ocean processes were responsible for a faster deep ocean circulation in the Pacific during glacial times. This is an important finding. The overall study is well thought out and the manuscript is well written.

I do have some comments and suggestions that are enumerated below.

Comments:

Working with Nd isotopes in the Pacific has always been a challenge since the water masses in the Pacific are not as distinct (in terms of Nd isotopic signature) as in the Atlantic. On the top of that, Pacific is full of volcanic materials and more often than we would like, these volcanic materials affect the water mass Nd isotopic signature and derail its water mass tracing property. More recently, a series of papers off Oregon have shown that pore water flux could have equally affect the bottom water values.

The main interpretation in this manuscript is contingent on demonstrating that Nd isotopes from Fe-Mn coating off foraminifera are pristine bottom water signal. The authors analyzed contemporaneous sediment samples from the same core as the foraminifera derived records to establish that non-seawater sources did not control the foraminifera derived Nd isotopic values. They have reported detrital values for two cores V21-30 and ODP 1241, both from equatorial Pacific and argued that patterns of changes in the detrital values are independent of foraminifera derived record. The authors pointed out that the detrital materials did not change in accordance with G-IG cycles and rather speculated the role of moving ITCZ as potential sources of external Nd. The ITCZ is also supposed to move on a G-IG cycle, so I am not sure what exactly the authors wanted to say here? So it is necessary that they clarify this a bit more. Also, I am curious

how studies of past ITCZ shifts (Jacobel et al., 2016) from the Pacific compare to their detrital record?

The detrital record of ODP 1241 is problematic specifically the last deglacial part. To me it appears that the detrital and foraminifera derived records follow the general pattern of change during the last deglacial. I would be very careful to claim that this part of the record is contamination free. If the foraminifera derived Nd isotopes following the last deglacial indeed represent contamination from detrital sediment, it becomes increasingly difficult to argue that the older part of the record is pristine seawater. The authors should address this issue in further detail. Moreover, ODP 1241 core top foraminifera Nd isotopic signature is 0.35, quite different from nearby modern seawater value at comparable depth (Grasse et al., 2012; Stn. 160). Wouldn't you expect seawater Nd isotopes to agree with core top foraminifera value?

The statement about porewater not affecting the bottom signal is well intentioned but it needs to be backed up with evidence. At this point it is a statement with some circumstantial evidence. While I am not sure how much of this is possible within the scope of the work presented here, it is worth the time to try a simple box model to show how much contribution from porewater is needed in order to create the downcore seawater Nd isotopic pattern that is observed in their record. If the model derived % contribution from porewater is anomalously high, it might be grounds to disregard the porewater contribution. Parameterization of such a model can be done using the published numbers off Oregon.

The Chatham Rise core CHAT16K does not show a lot of variations for the last 160 ka. The authors claimed that it is due to mixing between Tasman AAIW and southward circulation NPDW. It can also be argued to be modification due to local exchange (given how close it is to the continents and volcanic sources); can the authors strengthen their argument?

Minor comments:

Figure 5b is almost impossible to follow. Too busy.

Figure 6: Needs legends.

In summary, the approach taken by the authors to derived the transit time of deep water in the Pacific is new and quite interesting. While the modeling component is simple, it is a good start and hopefully there will be more systematic modeling attempts in the future.

Reference:

A. W. Jacobel, J. F. McManus, R. F. Anderson, G. Winckler, Large deglacial shifts of the Pacific Intertropical Convergence Zone. *Nat. Commun.* 7, 10449 (2016).

Reviewer #3 (Remarks to the Author):

Hu and Piotrowski use a combination of compiled and new Nd isotope records from foraminifera throughout the Pacific Basin to develop records of seawater Nd isotopic evolution over for the past 160 ky. This is augmented by detrital Nd isotope data from one region in the EEP. They use these data in a simplified model to argue for more rapid circulation in the Pacific during glacial times. This result has implications for understanding ocean processes on glacial-interglacial timescales as well carbon storage in the glacial ocean. There are some interesting aspects of the manuscript that make it appropriate for Nature Communications, but there are also aspects that require clarification and I worry it may be a bit of a challenge to address the changes I recommend below in a short journal format.

The authors do a nice job of setting up the importance of their work in the abstract and Introduction, but they could do a better job of summarizing the implications of their results within the framework of that set up. This might be accomplished effectively with a well-constructed Conclusion section. Some of that information is in the last section, but it would be helpful to the reader to wrap up the major implications succinctly. Along those lines, there could also be a comment at the end of the abstract to explain the implication of the statement that the glacial Pacific did not operate in a manner similar to the glacial Atlantic in terms of carbon storage.

The manuscript is organized in a reasonable fashion, but I also think the second author could give the manuscript a good read through to catch several grammatical errors. Without page, section or line numbers it is difficult to refer to specific aspects of the text. For this reason I have added minor comments to the word version of the text (attached) rather than at the end of this review. I also numbered the pages starting with the introduction and will refer to page numbers that way. Due to the addition of my comments, my page numbers may not match those in the original text.

In order to reach their conclusion about rapid glacial circulation in the Pacific, Hu and Piotrowski need to demonstrate that 1) glacial-interglacial (gl/ig) variations in ϵ_{Nd} of Pacific foraminifera are not driven by variations in detrital inputs and subsequent reactions within the sediment, in other words, the water chemistry is changing, not the detrital inputs and modification, and 2) the cause of the gl/ig isotopic variations in the water is caused by changes in the residence time of the water rather than changes in the ϵ_{Nd} value or flux from detrital sources that react with the waters along the flow path. At some level these two points are very similar, but they have chosen to address point one through a discussion of measured detrital ϵ_{Nd} values from one region, and point #2 through modeling.

One of my concerns relative to point 1 is that the flux of sediment and associated impacts on porewater chemistry and surface exposure time might influence the foraminiferal Nd isotopes. The detrital section (pg 9-10) is an

attempt to argue that the detrital inputs are not the cause of the changes observed in water chemistry. They appear to assume that the “equilibrium time” can be equated with the age of the sample, but perhaps it has more to do with time of foram exposure to seawater (sedimentation rate) or reactivity of the sediment. Pure carbonate sediments are likely to have very low porewater [Nd], but the addition of detrital material could significantly increase that concentration, as Abbott et al. (2015) showed. In addition, ash is likely to be more reactive than other continental sources (references by Wilson). In other words, there are clearly changes in seawater eNd on gl/ig timescales, but there are also likely to be changes in sedimentation. How can they eliminate the sedimentation changes as the source of the eNd variations? They could strengthen their argument for a seawater source of Nd in the ‘Foraminiferal versus detrital Nd isotope compositions’ section substantially by expanding their discussion on dust sources and providing some context for the magnitude of inputs through time. Additional detrital data from a region other than the EEP, which is near the end of the circulation cell, would also be useful, but time consuming to obtain. Given the data they have, it would help to describe potential detrital endmembers more broadly rather than simply focusing on EEP sources, and to be more precise in their discussion of EEP sources, which I found rather confusing. Hu and Piotrowski imply the only detrital source in this region is arc material; however, they also discuss riverine material, but then state that also has an arc composition. I was confused about the distinction. They play with variations in endmember compositions, but don’t clearly define the processes leading to those variations. Why wouldn’t loess inputs also increase during dustier glacial times? What is the distinction between ‘volcanic’ and ‘riverine’ arc inputs?

A more effective evaluation of the detrital contributions might include determining the percent detrital material (which might be equivalent to the inverse of % CaCO₃?) and sedimentation rate for each sample to see if there is a relationship between the foram-detrital Nd isotope offset and the amount of detrital material or exposure time; although, Wilson, from this research group, has pointed out that the composition of the detrital material also affects its reactivity. The general increase in foram eNd with increasing detrital eNd during terminations is what catches the eye in Figs. 3 and 4, and if they could demonstrate that the foram response isn’t linked through an increase in the rate of detrital inputs or the reactivity of the detrital inputs, it would strengthen their rather weak argument that “the evolution of bottom water Nd isotopes rather than detrital contamination has driven the foraminiferal εNd variability in our records.” I recognize that this is a short format journal, but I think much of this discussion could replace the existing discussion rather than simply building on it.

I also had a difficult time following the discussion for the N versus S regime patterns, starting with the discussion on pg 14. In order to see the patterns they present, it would be very helpful to label the sites and depths for each line

(perhaps in the appropriate color) for Figs. 5 and 6 because it is difficult to make out the subtle color differences in the short lines of the legend. It would also be a good idea to make the spacing along the Y axes the same for panels a and b to make comparisons easier. The point about higher ϵ_{Nd} values during glacials (less NACW) than interglacials is easy to see in the plots and makes sense, but I could not follow the discussion about the 'vertical convergence' at gl-ig transitions. I can see that there is less stratification during the glacials (which is most apparent in Fig 7c), but I don't understand the point they are trying to make about the gl-ig transitions, and Fig 7c makes it clear that the transitions are not the time of minimum offset. The next step in the discussion about decreased modification of ϵ_{Nd} during northward flow makes sense as an explanation for the minor reduction in the intermediate to deep offset in the North Pacific seen in glacials, but that decrease is very subtle and seems to vary from glacial to glacial, such that MIS 6 does not appear to reflect the decrease at all. Is there a way to highlight the offset for the reader, such as a plot of the difference between the two records? Initially, I assumed that the changes in the offset between northern and southern regimes was a function of whether or not that southern waters influenced by NACW were actually flowing north, thus I think they could state more clearly that they are arguing LCDW is flowing north throughout the record, but the seawater ϵ_{Nd} is modified more during slower transport during ig times than during the rapid transport during gl times. This statement is definitely in their text but it took a few readings for me to get there.

I do not do a lot of modeling and I'm not sure I exactly followed all of their description of what they did in the model, but it appears to be a very simplified view of the system. I have no objection to this type of simplification as long as the assumptions are clearly stated and justified, and the simplifications lead to robust results with interesting implications. One of my concerns about the short format for this paper is that I don't think the assumptions should be buried in the supplement. It's fine to put the details in the supplement and only include a summary of the results in the figures in the paper (as in Fig.7), but the reader needs enough information to evaluate the interpretations, and that is not the case in the current version of the manuscript. I think it is problematic to have the statement that "This model is a very simplified scheme which does not differentiate various sources of external Nd inputs with different Nd isotope compositions and fluxes since the main purpose for this model is to estimate the deep Pacific transit time instead of Nd cycling process" in the supplement and then the statement that "A simple mass balance calculation combined with paleoceanographic observations suggest the seawater ϵ_{Nd} cannot be explained by changes in the external Nd input alone" within the text. It seems that the first statement is telling me that they can't confidently evaluate the point they make in the second statement. Is it more accurate to simply say that the model illustrates that more rapid overturning in the glacial could produce their observations? Can they provide more information about the assumptions and outcomes of the model with reference to variations in detrital inputs to justify

their statement that the sediment modification of the seawater ϵ_{Nd} alone could not produce the observed pattern? In order to do that, I think they need to explain more clearly how the sediment might vary over gl/ig, and preferably back that up with some data.

In summary, I think Hu and Piotrowski have reached an interesting conclusion about variations in circulation in the Pacific during gl/ig that has important implications for understanding the carbon cycle and impacts of climate change. Thus, their results are appropriate for Nature Communications; however, they need to clarify a few points related to the potential for changes in sedimentation on gl/ig timescales to drive the seawater variations as noted above and provide more of an explanation of the model, particularly the assumptions that go into the simplification of the system, in the main text. If they can do that within the constraints of the paper limit, I would support publication in Nature Communications. Note, specific comments on wording and figures are included in track changes in the attachment.

Response to Reviewers' comments

- Reviewers' comments
- Authors' response: All the line numbers refer to the final revisions of the Manuscript and Supplementary Information.

We are very grateful to all the three reviewers for their helpful and constructive comments. We have endeavoured to make all changes suggested by the reviewers below. We believe this has produced a paper with more thorough discussion and more robust conclusions of our results.

Reviewer #1 was mostly supportive, and the comments are replied one-by-one in Reply 1-3. Both Reviewer #2 and #3 have suggested to clarify whether our foraminiferal Nd isotope records could reflect changing input of Nd, either by detrital source changes or through detrital dissolution in porewater. This issue will be addressed first by a detailed discussion about the potential detrital sources of our EEP cores, and then a quantitative estimation that shows the potential detrital influence on porewater Nd isotopic signatures is not likely to drive the foraminiferal ϵ_{Nd} variabilities. We will briefly summarize below:

Revision Summary 1: Detrital sources of our EEP cores

Previous work suggests that today the most important lithogenic inputs to the Panama Basin originate from the nearby central American Arcs¹ and Galapagos hotspot², which has been confirmed by the very positive detrital ϵ_{Nd} values of V21-30 (Fig.2c) and ODP1241 (Fig.3c) at the late Holocene in this study. The less radiogenic glacial detrital ϵ_{Nd} was not only found in the Panama Basin, but also in the Philippine Sea³, central⁴ and eastern⁵ equatorial Pacific regions, in accordance with higher Northern Hemisphere continental dust loads during the ice ages than during interglacials^{6, 7}. Unlike the western and central North Pacific, where most of the dust deposited originates in Asia^{3, 4, 8, 9}, the dust provenance of EEP might be different. Previous studies suggest its unradiogenic particles could also come from South America and Africa^{5, 10}.

Despite the complexity of detrital sources in the EEP, we divide them into two endmembers from the perspective of Nd isotopes: one from young volcanic arcs with radiogenic ϵ_{Nd} signatures and the other from old continental particles with unradiogenic ϵ_{Nd} signatures. The radiogenic volcanic component is likely from Central America arcs and Galapagos hotspot with average $\epsilon_{Nd} = +7^{ref11}$, while the unradiogenic component could be continental dust from Asia, South America and/or Africa with average $\epsilon_{Nd} = -10^{ref8}$. The variation in detrital ϵ_{Nd} of V21-30 and ODP1241 can thus be regarded as reflecting changing proportions of inputs from these two endmembers. Please see also Line 153-165 in Manuscript.

Revision Summary 2: Detrital influence on porewater ϵ_{Nd} record

The question of whether the shape and magnitude of change in the

foraminiferal ϵ_{Nd} records could have been actually caused by detrital contamination had been initially addressed in our initial submission based on the observation of different evolution patterns between foraminiferal and detrital ϵ_{Nd} records. However, we strengthen this argument here by quantitatively estimating the potential detrital influence on porewater Nd isotopic signatures considering the sedimentation changes via a binary mixing model¹²: the ϵ_{Nd} of the sedimentary porewater (ϵ_{Nd-pw}) is determined by a mixing of seawater derived Nd (authigenic) and lithogenic particulate Nd (old continental dust and young volcanic arc material). Please refer to Line 2-68 in Supplementary Method for details and Line 174-185 in Manuscript for calculation result.

Our calculation shows the modelled ϵ_{Nd-pw} G-I change from detrital contribution can explain only one third of the actual measured foraminiferal ϵ_{Nd} G-I change in our parameter setting (Supplementary Figs. 1, 2), which is barely analytically significant. The magnitude, timing and direction of change in the foraminiferal ϵ_{Nd} records still cannot be produced in the ϵ_{Nd-pw} records even when we force the detrital contamination to match the core-top and LGM foraminiferal ϵ_{Nd} values (Supplementary Fig. 3). Moreover, the close match between the foraminiferal ϵ_{Nd} records to benthic $\delta^{18}O$ (Fig. 2, 3), which is a proxy for deep water temperature and global ice volume, neither of which match the shifts in the detrital ϵ_{Nd} records, is also supportive of a global rather than local control on our foraminiferal ϵ_{Nd} records. We thus suggest that the foraminiferal ϵ_{Nd} variability in our records mainly reflect evolution of bottom water Nd isotopes.

Influence of globally increased glacial dust input with unradiogenic ϵ_{Nd} on Pacific seawater Nd isotopic composition is also considered as Reviewer #3 mentioned. We think the influence should be insignificant. First, modern Pacific seawater ϵ_{Nd} observation indicates minimal contribution of Nd originated from eolian dust¹³⁻¹⁵. Second, global Nd budget models^{43, 44} find a 2 to 3-fold increase in the Northern Hemisphere dust loads during the ice ages⁶ would not affect seawater ϵ_{Nd} in the Pacific Ocean^{16, 17}. Please see also Line 262-265 in Manuscript.

Reviewer #1 (Remarks to the Author):

Review of Hu & Piotrowski "Neodymium isotope evidence for glacial-interglacial variability of deepwater transit time in the Pacific Ocean", submitted to Nature Communications.

The paper presents a number of new Nd records from the deep Pacific in order to constrain orbital-scale changes in the Pacific overturning circulation across the last glacial-interglacial climate cycle (including isotope stages 1 to 6). The overturning circulation in the Pacific is one of the large unknowns in our knowledge of Quaternary paleoceanographic changes and their relation to potential storage of atmospheric CO₂. As such the new records by Hu & Piotrowski clearly merit publication in a high impact journal such as Nature

Communications.

The different geochemical records (seven novel and ten previously published sediment cores) are mainly from the South Pacific and the tropical sector of the Pacific. There are no records from the North Pacific itself (which could be mentioned somewhere in the text). The authors discuss the different records in quite some detail focussing on the main orbital trends, i.e. the glacial-interglacial variations. In this context, they distinguish a S-Pac versus N-Pacific regime. The southern one is characterized by more radiogenic ϵNd during glacials interpreted as reduced North Atlantic component water consistent with previous findings. From this, given the modern circulation strength, a more radiogenic glacial ϵNd signature would be expected for the North Pacific. In contrast, the records indicate similar or less radiogenic values in the tropical Pacific and shallower South Pacific return low (the N-Pacific regime). I think these conclusions are well documented and supported by the data. The only pitfall might be, that records from North Pacific itself are not included in the core collection used by the authors.

Reply 1: We thank Review#1 for the very supportive review of our manuscript. In this study, to avoid operational bias, the authigenic Nd isotope records in the Pacific compared are processed under a uniform technique, i.e. planktonic foraminifera with Fe-Mn coatings for consistency, but unfortunately there are no such records in the open North Pacific probably due to the extremely low sedimentation rates and poor preservation of carbonate. This is now mentioned in revised manuscript in Line 232.

The very important next step which makes this paper really suitable for a more general readership, is the transfer of the ϵNd signature changes into quantitative estimates, in this case deep Pacific transit time of water masses over the last 160 kyr. Hu & Piotrowski use a box model in order to estimate these transit times. The model is described in the supplementary material – a detailed evaluation of the suitability of this model is unfortunately beyond my expertise. In any case, the overall results are very important and provide evidence against the commonly assumed reduced Pacific overturning circulation during glacials. Instead, the reduced spatial ϵNd gradients and the box model results indicate substantially enhanced Pacific transect times during glacials, especially MIS2. If true, this finding has important implications for our mechanistic understanding of atmospheric CO_2 drawdown during glacials.

The paper is generally very precisely written. The results are well presented and the discussion is generally fine. One might add some additional records for deep Pacific circulation (e.g. proxies other than Nd) to a kind of summary figure. Some of those are discussed in the text but are not really shown in any figure (final chapter of the paper).

Reply 2: Other circulation proxies such as $\Delta^{14}\text{C}$ have similarly poor coverage due to low sedimentation rates and poor preservation of carbonate, and Cd/Ca

is poorly understood and may be affected by dissolution. However, similarly located and highly cited $\delta^{13}\text{C}$ record as a $\delta^{13}\text{C}$ gradient is included in Fig.7 which corroborates our interpretation based on Nd isotope records. See Line 323-331 in Manuscript.

A minor remark regarding Figure 1: I would split the zonal section into an eastern and western Pacific meridional section and may be a northern tropical zonal one.

Reply 3: The zonal section of Figure 1 is now split into a western (Fig.1b) and eastern (Fig.1c) Pacific meridional section showing the locations of all cores used in this study. See Line 78-91 in Manuscript.

Taken together, I think the extensive and novel Nd records and very challenging interpretation presented in this manuscript merits publication in Nature Communication.

Reviewer #2 (Remarks to the Author):

Hu and Piotrowski's 'Neodymium isotopic evidence for glacial-interglacial variability of deepwater transit time in the Pacific Ocean' is an interesting manuscript where the authors have taken an approach (rather a new approach in my view) to interpret Nd isotopes to derive transit time of deepwater in the Pacific. Their conclusion is interesting in the sense they find the Pacific to be more dynamic during glacial periods compared to the Atlantic. They hypothesized that high latitude Southern Ocean processes were responsible for a faster deep ocean circulation in the Pacific during glacial times. This is an important finding. The overall study is well thought out and the manuscript is well written.

I do have some comments and suggestions that are enumerated below.

Comments:

Working with Nd isotopes in the Pacific has always been a challenge since the water masses in the Pacific are not as distinct (in terms of Nd isotopic signature) as in the Atlantic. On the top of that, Pacific is full of volcanic materials and more often than we would like, these volcanic materials affect the water mass Nd isotopic signature and derail its water mass tracing property. More recently, a series of papers off Oregon have shown that pore water flux could have equally affect the bottom water values.

The main interpretation in this manuscript is contingent on demonstrating that Nd isotopes from Fe-Mn coating off foraminifera are pristine bottom water signal. The authors analyzed contemporaneous sediment samples from the same core as the foraminifera derived records to establish that non-seawater sources did not control the foraminifera derived Nd isotopic values. They have reported

detrital values for two cores V21-30 and ODP 1241, both from equatorial Pacific and argued that patterns of changes in the detrital values are independent of foraminifera derived record. The authors pointed out that the detrital materials did not change in accordance with G-IG cycles and rather speculated the role of moving ITCZ as potential sources of external Nd. The ITCZ is also supposed to move on a G-IG cycle, so I am not sure what exactly the authors wanted to say here? So it is necessary that they clarify this a bit more. Also, I am curious how studies of past ITCZ shifts (Jacobel et al., 2016) from the Pacific compare to their detrital record?

Reply 4: We thank Reviewer #2 for recognizing the novelty and importance of our work as well as raising some constructive comments.

Sorry for the obscurity of the discussion about ITCZ movement regarding to the changes in detrital sources in our initial manuscript. The detailed discussion about the detrital sources in the equatorial Pacific can be referred to Revision Summary 1, which is also included in Manuscript Line 153-165. The detrital sources of our cores are divided into two endmembers from the perspective of Nd isotopes: one from young volcanic arcs with radiogenic ϵ_{Nd} signatures (including Central America arcs and Galapagos hotspot) and the other from old continental crusts with unradiogenic ϵ_{Nd} signatures (such as Asia, South America and/or Africa).

A southward shift of the ITCZ during the LGM^{18, 19} can bring the southerly transport of Northern Hemisphere dust to the north of it and can also regulate the proximal source from the Central American continents via precipitation. But how ITCZ will influence the detrital sources of our cores is still unclear. With only two cores located at different depths above ~2 km exhibiting almost exactly the same values and evolution patterns in detrital ϵ_{Nd} records, it is hard to tie the observed spatial variability to past ITCZ movement. Such differentiation is also beyond the scope of this study and not closely relevant to our topic. So we have removed the statement about ITCZ movement in our revision.

The detrital record of ODP 1241 is problematic specifically the last deglacial part. To me it appears that the detrital and foraminifera derived records follow the general pattern of change during the last deglacial. I would be very careful to claim that this part of the record is contamination free. If the foraminifera derived Nd isotopes following the last deglacial indeed represent contamination from detrital sediment, it becomes increasingly difficult to argue that the older part of the record is pristine seawater. The authors should address this issue in further detail. Moreover, ODP 1241 core top foraminifera Nd isotopic signature is 0.35, quite different from nearby modern seawater value at comparable depth (Grasse et al., 2012; Stn. 160). Wouldn't you expect seawater Nd isotopes to agree with core top foraminifera value?

Reply 5: The seemingly similar evolution patterns of foraminiferal and detrital Nd isotopic records of our EEP cores (V21-30 and ODP1241) towards higher

ϵ_{Nd} values during the last deglacial leads to a question about whether the variation of foraminiferal ϵ_{Nd} records could be driven by detrital contamination in the porewater. Our model calculation suggests the observed variability in foraminiferal ϵ_{Nd} of V21-30 and ODP1241 over the studied periods, in magnitude, timing and direction, cannot be generated solely by detrital or porewater changes. Please refer to Revision Summary 2, and also see Line 174-185 in the Manuscript, Line 2-68 in the Supplementary Method as well as Supplementary Figs.1-3.

The statement about porewater not affecting the bottom signal is well intentioned but it needs to be backed up with evidence. At this point it is a statement with some circumstantial evidence. While I am not sure how much of this is possible within the scope of the work presented here, it is worth the time to try a simple box model to show how much contribution from porewater is needed in order to create the downcore seawater Nd isotopic pattern that is observed in their record. If the model derived % contribution from porewater is anomalously high, it might be grounds to disregard the porewater contribution. Parameterization of such a model can be done using the published numbers off Oregon.

Reply 6: Research has found that bottom water ϵ_{Nd} can be altered along the sediment-water boundary on the Oregon continental shelf⁴⁸. This in itself is not surprising, because the Oregon margin is a quite unique marine setting, with extremely high particle flux in the water column and very reducing environment in the sediment pore water, as evidenced by ferruginous pore fluids (50-250 μM on the shelf and 10-70 μM on the slope) and very shallow appearance of dissolved Fe (<15 cm)²⁰. In favour of lithogenic dissolution in the sediments, the lithogenic Nd flux on the Oregon margin²⁰ is around 3 orders of magnitude higher than the mean Pacific authigenic flux ($F_{\text{mean}} = 2.76 \times 10^{-5} \text{ g m}^{-2} \text{ yr}^{-1}$), and is therefore large enough to obscure the seawater Nd isotope composition²¹.

In contrast, the pore fluids in our EEP sites are unlikely to have experienced such reducing conditions because they have very low dissolved Fe (0-6.6 μM)² accompanied with abundant sulphates (>20 mM) within ~400 m below the seawater-sediment interface²². In this case, the parameterization inferred from off Oregon are probably not appropriate for our EEP sites, and a quantitative examination of the pore water contribution to seawater Nd isotope evolution is not likely without suitable porewater parameterization. Moreover, our model calculation also denied the dominant role of Nd contamination from dissolving detritus in driving the foraminiferal Nd isotope variations on G-I cycles.

The Chatham Rise core CHAT16K does not show a lot of variations for the last 160 ka. The authors claimed that it is due to mixing between Tasman AAIW and southward circulation NPDW. It can also be argued to be modification due to local exchange (given how close it is to the continents and volcanic sources); can the authors strengthen their argument?

Reply 7: We agree that modification due to local exchange with volcanic particulates derived from New Zealand's North islands may have occurred on Chatham Rise²³. But without modern seawater ϵ_{Nd} measurements around CHAT16K, it is difficult to distinguish between the local exchange effect and water mass mixing²⁴. At this point, neither scenarios can be excluded. See Line 243-248 in Manuscript for revision.

Minor comments:

Figure 5b is almost impossible to follow. Too busy.

Figure 6: Needs legends.

Reply 8: Each core name and depth has been added along the corresponding record using the same colour in Fig.5 and Fig.6.

In summary, the approach taken by the authors to derive the transit time of deep water in the Pacific is new and quite interesting. While the modeling component is simple, it is a good start and hopefully there will be more systematic modeling attempts in the future.

Reviewer #3 (Remarks to the Author):

Hu and Piotrowski use a combination of compiled and new Nd isotope records from foraminifera throughout the Pacific Basin to develop records of seawater Nd isotopic evolution over for the past 160 ky. This is augmented by detrital Nd isotope data from one region in the EEP. They use these data in a simplified model to argue for more rapid circulation in the Pacific during glacial times. This result has implications for understanding ocean processes on glacial-interglacial timescales as well carbon storage in the glacial ocean. There are some interesting aspects of the manuscript that make it appropriate for Nature Communications, but there are also aspects that require clarification and I worry it may be a bit of a challenge to address the changes I recommend below in a short journal format.

The authors do a nice job of setting up the importance of their work in the abstract and Introduction, but they could do a better job of summarizing the implications of their results within the framework of that set up. This might be accomplished effectively with a well-constructed Conclusion section. Some of that information is in the last section, but it would be helpful to the reader to wrap up the major implications succinctly. Along those lines, there could also be a comment at the end of the abstract to explain the implication of the statement that the glacial Pacific did not operate in a manner similar to the glacial Atlantic in terms of carbon storage.

Reply 9: We thank Reviewer #3 for recognizing the importance of our work and constructive comments. Since the journal of *Nature Communications* usually

does not have a Conclusion section, we wrap up the implication of our findings by synthesizing the hypothetical Pacific overturning circulation states during the glacial and interglacial states in our final chapter (see Line 332-336) and final Fig.8 (Line 357-368). At the end of the final chapter, the conclusion sentences for glacial overturning circulation state and potential associated mechanisms as well as its implication for future climate models are revised (See Line 360-356). An implication sentence is now added at the end of the abstract (see Line 18-21).

The manuscript is organized in a reasonable fashion, but I also think the second author could give the manuscript a good read through to catch several grammatical errors. Without page, section or line numbers it is difficult to refer to specific aspects of the text. For this reason I have added minor comments to the word version of the text (attached) rather than at the end of this review. I also numbered the pages starting with the introduction and will refer to page numbers that way. Due to the addition of my comments, my page numbers may not match those in the original text.

Reply 10: All the minor comments in the attachment has been revised according to the attached pdf.

In order to reach their conclusion about rapid glacial circulation in the Pacific, Hu and Piotrowski need to demonstrate that 1) glacial-interglacial (gl/ig) variations in eNd of Pacific foraminifera are not driven by variations in detrital inputs and subsequent reactions within the sediment, in other words, the water chemistry is changing, not the detrital inputs and modification, and 2) the cause of the gl/ig isotopic variations in the water is caused by changes in the residence time of the water rather than changes in the eNd value or flux from detrital sources that react with the waters along the flow path. At some level these two points are very similar, but they have chosen to address point one through a discussion of measured detrital eNd values from one region, and point #2 through modeling.

One of my concerns relative to point 1 is that the flux of sediment and associated impacts on porewater chemistry and surface exposure time might influence the foraminiferal Nd isotopes. The detrital section (pg 9-10) is an attempt to argue that the detrital inputs are not the cause of the changes observed in water chemistry. They appear to assume that the “equilibrium time” can be equated with the age of the sample, but perhaps it has more to do with time of foram exposure to seawater (sedimentation rate) or reactivity of the sediment. Pure carbonate sediments are likely to have very low porewater [Nd], but the addition of detrital material could significantly increase that concentration, as Abbott et al. (2015) showed. In addition, ash is likely to be more reactive than other continental sources (references by Wilson). In other words, there are clearly changes in seawater eNd on gl/ig timescales, but there are also likely to be changes in sedimentation. How can they eliminate the

sedimentation changes as the source of the ϵ_{Nd} variations?

They could strengthen their argument for a seawater source of Nd in the 'Foraminiferal versus detrital Nd isotope compositions' section substantially by expanding their discussion on dust sources and providing some context for the magnitude of inputs through time. Additional detrital data from a region other than the EEP, which is near the end of the circulation cell, would also be useful, but time consuming to obtain. Given the data they have, it would help to describe potential detrital endmembers more broadly rather than simply focusing on EEP sources, and to be more precise in their discussion of EEP sources, which I found rather confusing. Hu and Piotrowski imply the only detrital source in this region is arc material; however, they also discuss riverine material, but then state that also has an arc composition. I was confused about the distinction. They play with variations in endmember compositions, but don't clearly define the processes leading to those variations. Why wouldn't loess inputs also increase during dustier glacial times? What is the distinction between 'volcanic' and 'riverine' arc inputs?

A more effective evaluation of the detrital contributions might include determining the percent detrital material (which might be equivalent to the inverse of % CaCO₃?) and sedimentation rate for each sample to see if there is a relationship between the foram-detrital Nd isotope offset and the amount of detrital material or exposure time; although, Wilson, from this research group, has pointed out that the composition of the detrital material also affects its reactivity. The general increase in foram ϵ_{Nd} with increasing detrital ϵ_{Nd} during terminations is what catches the eye in Figs. 3 and 4, and if they could demonstrate that the foram response isn't linked through an increase in the rate of detrital inputs or the reactivity of the detrital inputs, it would strengthen their rather weak argument that "the evolution of bottom water Nd isotopes rather than detrital contamination has driven the foraminiferal ϵ_{Nd} variability in our records." I recognize that this is a short format journal, but I think much of this discussion could replace the existing discussion rather than simply building on it.

Reply 11: We appreciate the suggestion of using both the detrital flux and sedimentation rate to evaluate the detrital contribution of our EEP cores ODP1241 and V21-30 and agree that an increase in the rate of detrital inputs or the reactivity of the detrital inputs has the potential to change the porewater Nd isotopic composition as evidenced in our porewater ϵ_{Nd} calculations (Supplementary Fig.1-2). We thank Reviewer #3 for inspiring us making a quantitative estimation based on the above idea. The modelled ϵ_{Nd-pw} records share quite a few similarities with the detrital ϵ_{Nd} evolution patterns and show more radiogenic values when sedimentation rates peaked, confirming the influence of changing sedimentation, but could not produce the variabilities of our foraminiferal ϵ_{Nd} records. Please refer to Revision Summary 2, and also see Line 174-185 in the Manuscript, Line 2-68 in the Supplementary Method as well as Supplementary Figs.1-3 for details.

Regarding the detrital sources of our EEP cores, we divide them into two endmembers from the perspective of Nd isotopes: one from young volcanic arcs with radiogenic ϵ_{Nd} signatures and the other from old continental crusts with unradiogenic ϵ_{Nd} signatures. The consistency in increased detrital ϵ_{Nd} during the last deglacial period over the entire equatorial Pacific indicates a decrease of old-continental dust input with unradiogenic ϵ_{Nd} compositions to the deep-sea sediments since the LGM^{4,5}, in agreement with dustier glacial times^{6,7}. Although the drivers of increased glacial dust flux are beyond the scope of our study, previous work suggest increased in dust availability, entrainment and transport from these potential sources may have played a significant role for the ~2-3 times larger dust flux in the equatorial Pacific during the LGM^{19, 25-29}. For example, dryness in central America²⁸ could have provided a proximal dust source to the EEP. A southward shift of the ITCZ during the LGM¹⁸ would have placed ODP1241 and V21-30 to the north of it, exposing these locations to higher Northern Hemisphere dust loads. In addition, the slowdown of the AMOC appears to have caused massive drying in the northern Africa²⁹, which might have served as another distal source to the EEP. These potential sources are supported by dust-provenance studies^{5, 6, 8, 9}.

The influence of globally increased glacial dust input with unradiogenic ϵ_{Nd} on Pacific seawater Nd isotopic composition which should be insignificant is also included in Revision Summary 2 and Line 262-265 in Manuscript.

The volcanic arc materials are probably brought to the Panama Basin in two ways: 1) as river loads from large Colombian rivers originated from Central America Arc; 2) as ash or pumice through volcanic eruption of Galapagos hotspot and Central America. Although these two components have similar ϵ_{Nd} compositions¹¹, their difference in dissolution factors due to more reactive fresh volcanic ash than the weathered riverine particulate³⁰ would lead to discrepancy in the concentration of dissolved Nd from detritus in the porewater. Since it is impossible to quantitatively assess the contents of these two components at this stage, they are not differentiated in our model. Despite of that, we would not expect a significant difference in our conclusion. The risk of possible contamination from labile terrigenous fraction such as volcanogenic material is minimized for the foraminifera archive, because the detrital particles are physically removed from foraminiferal tests. This is supported by the absence in causal relationship between foraminiferal ϵ_{Nd} peaks and volcanic ash/lapilli layers of ODP1241^{2, 31} and CHAT5K³². Therefore, we demonstrate that the foraminiferal ϵ_{Nd} response is not linked through changes in the rate or the reactivity of the detrital inputs.

I also had a difficult time following the discussion for the N versus S regime patterns, starting with the discussion on pg 14. In order to see the patterns they present, it would be very helpful to label the sites and depths for each line (perhaps in the appropriate color) for Figs. 5 and 6 because it is difficult to make out the subtle color differences in the short lines of the legend. It would also be

a good idea to make the spacing along the Y axes the same for panels a and b to make comparisons easier. The point about higher ϵ_{Nd} values during glacials (less NACW) than interglacials is easy to see in the plots and makes sense, but I could not follow the discussion about the 'vertical convergence' at gl-ig transitions. I can see that there is less stratification during the glacials (which is most apparent in Fig 7c), but I don't understand the point they are trying to make about the gl-ig transitions, and Fig 7c makes it clear that the transitions are not the time of minimum offset. The next step in the discussion about decreased modification of ϵ_{Nd} during northward flow makes sense as an explanation for the minor reduction in the intermediate to deep offset in the North Pacific seen in glacials, but that decrease is very subtle and seems to vary from glacial to glacial, such that MIS 6 does not appear to reflect the decrease at all. Is there a way to highlight the offset for the reader, such as a plot of the difference between the two records? Initially, I assumed that the changes in the offset between northern and southern regimes was a function of whether or not that southern waters influenced by NACW were actually flowing north, thus I think they could state more clearly that they are arguing LCDW is flowing north throughout the record, but the seawater ϵ_{Nd} is modified more during slower transport during ig times than during the rapid transport during gl times. This statement is definitely in their text but it took a few readings for me to get there.

Reply 12: Records in Fig.5 and Fig.6 have been labelled with core names and depths in matching colours respectively) and the same spacing of Y axes of (a) and (b) have been adjusted. See Line 272-280 in Manuscript.

The discussion about 'vertical convergence' at G-I transitions for the South Pacific records (Fig.5b, 6b, not the N-S deep ϵ_{Nd} gradient in Fig.7c) have been clarified in our previous work³². In that paper, we found diminished ϵ_{Nd} difference between deep records of CHAT1K/5K (water depth 3556-4220 m) and intermediate record of Y9 (water depth 1267 m) since deglaciation. But this is not a major point of this manuscript and thus has been removed (see Line 236-239 for the revision). The focus in this study is the spatial ϵ_{Nd} difference between South and North deep Pacific (ODP1123 and ODP846), which is plotted in Fig.7c, shows a decrease from ~4-5 ϵ_{Nd} units in MIS 1, 3 and 5 to ~2-3 ϵ_{Nd} units during MIS 2, 4 and 6 (Fig. 7c).

I do not do a lot of modeling and I'm not sure I exactly followed all of their description of what they did in the model, but it appears to be a very simplified view of the system. I have no objection to this type of simplification as long as the assumptions are clearly stated and justified, and the simplifications lead to robust results with interesting implications. One of my concerns about the short format for this paper is that I don't think the assumptions should be buried in the supplement. It's fine to put the details in the supplement and only include a summary of the results in the figures in the paper (as in Fig.7), but the reader needs enough information to evaluate the interpretations, and that is not the

case in the current version of the manuscript. I think it is problematic to have the statement that “This model is a very simplified scheme which does not differentiate various sources of external Nd inputs with different Nd isotope compositions and fluxes since the main purpose for this model is to estimate the deep Pacific transit time instead of Nd cycling process” in the supplement and then the statement that “A simple mass balance calculation combined with paleoceanographic observations suggest the seawater ϵ_{Nd} cannot be explained by changes in the external Nd input alone” within the text. It seems that the first statement is telling me that they can’t confidently evaluate the point they make in the second statement. Is it more accurate to simply say that the model illustrates that more rapid overturning in the glacial could produce their observations? Can they provide more information about the assumptions and outcomes of the model with reference to variations in detrital inputs to justify their statement that the sediment modification of the seawater ϵ_{Nd} alone could not produce the observed pattern? In order to do that, I think they need to explain more clearly how the sediment might vary over gl/ig, and preferably back that up with some data.

Reply 13: We agree that both the assumptions and results of the model should be clearly stated in the main text. Model description now can be found in the Method section in the Manuscript (Line 421-446), where the deep Pacific Ocean is treated as a box with LCDW exclusively ventilated from the Southern Ocean, and the water mass ϵ_{Nd} signatures are assumed to be gradually modified in the process of northward LCDW advection by time-dependant accumulation of external Nd input from a constant source²⁴. All the parameters and their values adopted from literatures in the model are shown in Supplementary Table 10. The results are shown in Line 302-310 and Fig.7.

The detrital influence on foraminiferal ϵ_{Nd} records should be insignificant as discussed in Line 174-185 in Manuscript. Please also refer to Revision Summary 2, and Line 2-68 in Supplementary Method and Supplementary Figs.1-3. The potential porewater G-I ϵ_{Nd} change from detrital contribution can explain only one third of the actual measured foraminiferal ϵ_{Nd} G-I change (Supplementary Figs. 1, 2), which is barely analytically significant. The magnitude, timing and direction of change in the foraminiferal ϵ_{Nd} records still cannot be produced in the ϵ_{Nd-pw} records even when we force the detrital contamination to match the core-top and LGM foraminiferal ϵ_{Nd} values (Supplementary Fig. 3). Moreover, the close match between the foraminiferal ϵ_{Nd} records to benthic $\delta^{18}O$ (Fig. 2, 3), which is a proxy for deep water temperature and global ice volume, neither of which match the shifts in the detrital ϵ_{Nd} records, is also supportive of a global rather than local control on our foraminiferal ϵ_{Nd} records. We thus suggest that the foraminiferal ϵ_{Nd} variability in our records mainly reflect evolution of bottom water Nd isotopes.

In summary, I think Hu and Piotrowski have reached an interesting conclusion about variations in circulation in the Pacific during gl/ig that has important

implications for understanding the carbon cycle and impacts of climate change. Thus, their results are appropriate for Nature Communications; however, they need to clarify a few points related to the potential for changes in sedimentation on gl/ig timescales to drive the seawater variations as noted above and provide more of an explanation of the model, particularly the assumptions that go into the simplification of the system, in the main text. If they can do that within the constraints of the paper limit, I would support publication in Nature Communications. Note, specific comments on wording and figures are included in track changes in the attachment.

Reply 14: With proper arrangement of the contents we believe the revised contents could be nicely incorporated into the manuscript without making it too long. We thank reviewer #3 again for the detailed suggestions to our manuscript that helped us to clarify quite a few important issues.

References in Response

1. Grasse, P. *et al.* Short-term variability of dissolved rare earth elements and neodymium isotopes in the entire water column of the Panama Basin. *Earth Planet Sc Lett* **475**, 242-253 (2017).
2. Mix, A.C., Tiedemann, R., Baldauf, J. & Blum, P. Site 1241. Proceedings of the Ocean Drilling Program, Initial Report **202**, IR-12 (2003).
3. Jiang, F. *et al.* Asian dust input in the western Philippine Sea: Evidence from radiogenic Sr and Nd isotopes. *Geochemistry, Geophysics, Geosystems* **14**, 1538-1551 (2013).
4. Reimi, M.A. & Marcantonio, F. Constraints on the magnitude of the deglacial migration of the ITCZ in the Central Equatorial Pacific Ocean. *Earth Planet Sc Lett* **453**, 1-8 (2016).
5. Xie, R.C. & Marcantonio, F. Deglacial dust provenance changes in the Eastern Equatorial Pacific and implications for ITCZ movement. *Earth Planet Sc Lett* **317–318**, 386-395 (2012).
6. Winckler, G., Anderson, R.F., Fleisher, M.Q., McGee, D. & Mahowald, N. Covariant Glacial-Interglacial Dust Fluxes in the Equatorial Pacific and Antarctica. *Science* **320**, 93-96 (2008).
7. Anderson, R.F., Fleisher, M.Q. & Lao, Y. Glacial–interglacial variability in the delivery of dust to the central equatorial Pacific Ocean. *Earth Planet Sc Lett* **242**, 406-414 (2006).
8. Nakai, S.i., Halliday, A.N. & Rea, D.K. Provenance of dust in the Pacific Ocean. *Earth Planet Sc Lett* **119**, 143-157 (1993).
9. Stancin, A.M. *et al.* Radiogenic isotopic mapping of late Cenozoic eolian and hemipelagic sediment distribution in the east-central Pacific. *Earth Planet Sc Lett* **248**, 840-850 (2006).
10. Kienast, S.S., Kienast, M., Mix, A.C., Calvert, S.E. & François, R. Thorium-230 normalized particle flux and sediment focusing in the Panama Basin region during the last 30,000 years. *Paleoceanography* **22**, PA2213 (2007).
11. Grenier, M. *et al.* From the subtropics to the central equatorial Pacific Ocean: Neodymium isotopic composition and rare earth element concentration variations. *Journal of Geophysical Research: Oceans* **118**, 592-618 (2013).
12. Tachikawa, K., Handel, C. & Dupré, B. Distribution of rare earth elements and neodymium isotopes in settling particulate material of the tropical Atlantic Ocean (EUMELI site). *Deep Sea Research Part I: Oceanographic Research Papers* **44**, 1769-1792 (1997).
13. Jones, C.E., Halliday, A.N., Rea, D.K. & Owen, R.M. Neodymium isotopic variations in North Pacific modern silicate sediment and the insignificance of detrital REE contributions to seawater. *Earth Planet Sc Lett* **127**, 55-66 (1994).

14. Amakawa, H. *et al.* Neodymium isotopic variations in Northwest Pacific waters. *Geochim Cosmochim Acta* **68**, 715-727 (2004).
15. Lacan, F. & Jeandel, C. Tracing Papua New Guinea imprint on the central Equatorial Pacific Ocean using neodymium isotopic compositions and Rare Earth Element patterns. *Earth Planet Sc Lett* **186**, 497-512 (2001).
16. Rempfer, J., Stocker, T.F., Joos, F. & Dutay, J.-C. Sensitivity of Nd isotopic composition in seawater to changes in Nd sources and paleoceanographic implications. *J. Geophys. Res.* **117**, C12010 (2012).
17. Tachikawa, K., Athias, V. & Jeandel, C. Neodymium budget in the modern ocean and paleo-oceanographic implications. *J. Geophys. Res.* **108**, 3254 (2003).
18. Koutavas, A. & Lynch-Stieglitz, J. Glacial-interglacial dynamics of the eastern equatorial Pacific cold tongue-Intertropical Convergence Zone system reconstructed from oxygen isotope records. *Paleoceanography* **18**, 1089 (2003).
19. Jacobel, A.W., McManus, J.F., Anderson, R.F. & Winckler, G. Climate-related response of dust flux to the central equatorial Pacific over the past 150 kyr. *Earth Planet Sc Lett* **457**, 160-172 (2017).
20. Abbott, A.N., Haley, B.A., McManus, J. & Reimers, C.E. The sedimentary flux of dissolved rare earth elements to the ocean. *Geochim Cosmochim Acta* **154**, 186-200 (2015).
21. Tachikawa, K., Jeandel, C., Vangriesheim, A. & Dupré, B. Distribution of rare earth elements and neodymium isotopes in suspended particles of the tropical Atlantic Ocean (EUMELI site). *Deep Sea Research Part I: Oceanographic Research Papers* **46**, 733-755 (1999).
22. Mayer, L., Pisias, N., Janecek, T., *et al.* Site 846. Proceedings of the Ocean Drilling Program, Initial Report **138**, IR-11 (1992).
23. Noble, T.L., Piotrowski, A.M. & McCave, I.N. Neodymium isotopic composition of intermediate and deep waters in the glacial southwest Pacific. *Earth Planet Sc Lett* **384**, 27-36 (2013).
24. Hu, R., Piotrowski, A.M., Bostock, H.C., Crowhurst, S. & Rennie, V. Variability of neodymium isotopes associated with planktonic foraminifera in the Pacific Ocean during the Holocene and Last Glacial Maximum. *Earth Planet Sc Lett* **447**, 130-138 (2016).
25. Rea, D.K. The paleoclimatic record provided by eolian deposition in the deep sea: The geologic history of wind. *Reviews of Geophysics* **32**, 159-195 (1994).
26. McGee, D., Broecker, W.S. & Winckler, G. Gustiness: The driver of glacial dustiness? *Quaternary Science Reviews* **29**, 2340-2350 (2010).
27. Serno, S. *et al.* Change in dust seasonality as the primary driver for orbital-scale dust storm variability in East Asia. *Geophysical Research Letters* **44**, 3796-3805 (2017).
28. Hodell, D.A. *et al.* An 85-ka record of climate change in lowland Central America. *Quaternary Science Reviews* **27**, 1152-1165 (2008).
29. Mulitza, S. *et al.* Sahel megadroughts triggered by glacial slowdowns of Atlantic meridional overturning. *Paleoceanography* **23** (2008).
30. Jones, M.T. *et al.* Riverine particulate material dissolution as a significant flux of strontium to the oceans. *Earth Planet Sc Lett* **355–356**, 51-59 (2012).
31. Lalicata, J.J. & Lea, D.W. Pleistocene carbonate dissolution fluctuations in the eastern equatorial Pacific on glacial timescales: Evidence from ODP Hole 1241. *Marine Micropaleontology* **79**, 41-51 (2011).
32. Hu, R. *et al.* Neodymium isotopic evidence for linked changes in Southeast Atlantic and Southwest Pacific circulation over the last 200 kyr. *Earth Planet Sc Lett* **455**, 106-114 (2016).

Reviewers' comments:

Reviewer #1 (Remarks to the Author):

Review of Hu & Piotrowski "Neodymium isotope evidence for glacial-interglacial variability of deepwater transit time in the Pacific Ocean", submitted to Nature Communications.

The revised paper nicely addresses and incorporates the comments and suggestions of 3 reviewers. Though reviewers #2 and 3 were more critical, all reviewers were overall positive and recommended publication in Nature Communications after revision.

In my opinion the authors were successful in addressing the comments of the reviewers. My own comments were all recognized and addressed adequately. The same applies for the very detailed replies to the individual points of the other reviewers. The revisions done in the manuscripts clearly address these replies in the rebuttal letter.

Therefore, given the high quality of the data, and the novel interpretations, I clearly recommend publication in Nature Communications without further revisions.

Reviewer #2 (Remarks to the Author):

Hu and Piotrowski have done a good job revising their manuscript entitled "Neodymium isotope evidence for glacial-interglacial variability of deepwater transit time in the Pacific Ocean". The authors have addressed all comments and have now included a modeling component which in my opinion has strengthened their argument. This is an interesting contribution and I recommend acceptance for publication in Nature Communications.

I have a few minor editorial comments:

Line 26: Check sentence for wording.

Line 33: "more poorly" sounds over emphasizing.

Line 39: "intimately linked" sounds over emphasizing.

Line 75: "... how the strength of the overturning circulation 'is' linked to carbon storage..."

Fig.1: Please review how you describe the legends used in the figure. It is hard to follow and confusing.

Line:180: This is one long sentence. Re-phrasing the sentence will increase readability.

Reviewer #3 (Remarks to the Author):

Hu and Piotrowski followed the guidance provided by reviewers and made minor modifications the manuscript to clarify a number of points. The addition

of the “Examination of detrital influence on porewater eNd record” in the supplement and a summary of those results in the main text was particularly useful. [Note “Examination of detrital influence on porewater eNd records” would be more grammatically correct.]

In several cases it seemed that they addressed reviewer’s comments by throwing in a statement that was not really woven into the manuscript. One example of this is the addition of the sentence started with “A 2 to 3-fold...” on line 262. I understood the point of this statement based on reading reviewers’ comments, but not based on the text leading up to the inserted sentence. Sometimes the sentences added here or there did make a big difference, but other times a bit more effort would have helped to get the point across better.

Regardless, the manuscript presents interesting new data sets and the modeling is a good start at evaluating the processes driving changes in Nd isotopes. The key contributions of the article, such as the novel application of Nd isotopes to track circulation rates and the concept of more vigorous Pacific overturning circulation during the last glacial which has implications for carbon cycling on glacial/interglacial time scales, are interesting and worthy of publication in Nature Communications.

Response to Reviewers' comments

- Reviewers' comments
- Authors' response: All the line numbers refer to the final revisions of the Manuscript (Track change version).

We are very grateful to all three reviewers for their extremely constructive reviews throughout the review process. We have addressed their remaining concerns as outlined below.

Reviewer #1 (Remarks to the Author):

Review of Hu & Piotrowski "Neodymium isotope evidence for glacial-interglacial variability of deepwater transit time in the Pacific Ocean", submitted to Nature Communications.

The revised paper nicely addresses and incorporates the comments and suggestions of 3 reviewers. Though reviewers #2 and 3 were more critical, all reviewers were overall positive and recommended publication in Nature Communications after revision.

In my opinion the authors were successful in addressing the comments of the reviewers. My own comments were all recognized and addressed adequately. The same applies for the very detailed replies to the individual points of the other reviewers. The revisions done in the manuscripts clearly address these replies in the rebuttal letter.

Therefore, given the high quality of the data, and the novel interpretations, I clearly recommend publication in Nature Communications without further revisions.

Reply 1: Thank you.

Reviewer #2 (Remarks to the Author):

Hu and Piotrowski have done a good job revising their manuscript entitled "Neodymium isotope evidence for glacial-interglacial variability of deepwater transit time in the Pacific Ocean". The authors have addressed all comments and have now included a modeling component which in my opinion has strengthened their argument. This is an interesting contribution and I recommend acceptance for publication in Nature Communications.

I have a few minor editorial comments:

Line 26: Check sentence for wording.

Reply 2: See Line 26 for the re-wording of this sentence.

Line 33: “more poorly” sounds over emphasizing.

Reply 3: “more ... than in the Atlantic” has been deleted. See Line 33-34 for the revision.

Line 39: “intimately linked” sounds over emphasizing.

Reply 4: This sentence has been changed to “These proxies are also controlled by a combination of ...”. See Line 39-40 for the revision.

Line 75: “... how the strength of the overturning circulation ‘is’ linked to carbon storage...”

Reply 5: Changed. See Line 75 for the revision.

Fig.1: Please review how you describe the legends used in the figure. It is hard to follow and confusing.

Reply 6: The legends of Fig.1 are now revised. See Line 684-696 for the revision.

Line:180: This is one long sentence. Re-phrasing the sentence will increase readability.

Reply 7: This sentence has now been re-phrased. See Line 153-157 for the revision.

Reviewer #3 (Remarks to the Author):

Hu and Piotrowski followed the guidance provided by reviewers and made minor modifications the manuscript to clarify a number of points. The addition of the “Examination of detrital influence on porewater eNd record” in the supplement and a summary of those results in the main text was particularly useful. [Note “Examination of detrital influence on porewater eNd records” would be more grammatically correct.]

Reply 8: The subtitle has been revised accordingly and this part has now been moved to the Methods section in the main text. See Line 352 for the revision.

In several cases it seemed that they addressed reviewer’s comments by throwing in a statement that was not really woven into the manuscript. One

example of this is the addition of the sentence started with “A 2 to 3-fold...” on line 262. I understood the point of this statement based on reading reviewers’ comments, but not based on the text leading up to the inserted sentence. Sometimes the sentences added here or there did make a big difference, but other times a bit more effort would have helped to get the point across better.

Reply 9: To get across better, this point has now been moved to Line 221-224.

Regardless, the manuscript presents interesting new data sets and the modeling is a good start at evaluating the processes driving changes in Nd isotopes. The key contributions of the article, such as the novel application of Nd isotopes to track circulation rates and the concept of more vigorous Pacific overturning circulation during the last glacial which has implications for carbon cycling on glacial/interglacial time scales, are interesting and worthy of publication in Nature Communications.

Reply 10: Thank you.